# Is Finer Better? The Limits of Microscaling Formats in Large Language Models

**Andrea Fasoli**      **Monodeep Kar**      **Chi-Chun Liu**      **Swagath Venkataramani**

**Viji Srinivasan**      **Leland Chang**      **Naigang Wang**

IBM Research, USA

{andrea.fasoli, monodeep.kar, swagath.venkataramani}@ibm.com

{cliu, viji, lelandc, nwang}@us.ibm.com

## Abstract

Microscaling data formats leverage per-block tensor quantization to enable aggressive model compression with limited loss in accuracy. Unlocking their potential for efficient training and inference necessitates hardware-friendly implementations that handle matrix multiplications in a native format and adopt efficient error-mitigation strategies. Herein, we report the emergence of a surprising behavior associated with microscaling quantization, whereas the output of a quantized model degrades as block size is decreased below a given threshold. This behavior clashes with the expectation that a smaller block size should allow for a better representation of the tensor elements. We investigate this phenomenon both experimentally and theoretically, decoupling the sources of quantization error behind it. Experimentally, we analyze the distributions of several Large Language Models and identify the conditions driving the anomalous behavior. Theoretically, we lay down a framework showing remarkable agreement with experimental data from pretrained model distributions and ideal ones. Overall, we show that the anomaly is driven by the interplay between narrow tensor distributions and the limited dynamic range of the quantized scales. Based on these insights, we propose the use of FP8 unsigned E5M3 (UE5M3) as a novel hardware-friendly format for the scales in FP4 microscaling data types. We demonstrate that UE5M3 achieves comparable performance to the conventional FP8 unsigned E4M3 scales while obviating the need of global scaling operations on weights and activations.

## 1 Introduction

The unprecedented growth of large language models (LLMs) has brought dramatic improvements in natural language processing, but at the expense of escalating compute, memory, and energy demands (Kaplan et al. (2020),Hoffmann et al. (2022),Samsi et al. (2023)). With model sizes reaching hundreds of billions of parameters and context windows extending to hundreds of thousands of tokens, reducing numerical precision has emerged as a cornerstone for enabling efficient training and inference (Gupta et al. (2015),Kuzmin et al. (2022),Xiao et al. (2023)). Hardware vendors have progressively shifted from FP16 to FP8, and now to FP4, in order to increase throughput and energy efficiency. Yet, pushing precision below 8 bits often leads to substantial accuracy degradation, particularly when both weights and activations are quantized (Dettmers & Zettlemoyer (2023),Ashkboos et al. (2024),Li et al. (2025)).

Over the past few years, quantization techniques for LLMs have steadily evolved towards finer-grained control in order to balance accuracy and efficiency. Early approaches primarily employed tensor-wide quantization, assigning a single scale factor to an entire weight or activation tensor. However, this often introduced large quantization errors in regions with high dynamic range (Lin et al. (2020), Dettmers et al. (2022)). To address this limitation, per-channel and per-token quantization were introduced, assigning independent scales to individual output channels or tokens, and thereby achieving significant accuracy improvements at modest storage and compute costs (Dai et al. (2021),Yao et al. (2022)). More recently, per-group quantization has gained traction, wherein small

groups of elements (e.g., 128) share scaling factors, offering a compromise between reduced quantization error and the increased overhead due to the proliferation of quantization parameters (Shen et al. (2020),Frantar et al. (2022),Xiao et al. (2023)).

Microscaling formats (Rouhani et al. (2023a)) push this trend towards even finer quantization resolutions: by means of efficient hardware implementations coupled with scaling factor quantization, these formats enable grouping elements in smaller blocks, while limiting data transfer overheads. The benefits of this strategy has motivated rapid industry adoption, with microscaling formats now supported in commercial AI accelerators (NVIDIA (2025a); AMD (2025)). Despite these successes, the robustness of microscaling, particularly at sub-8-bit precision, remains an open question that is still subject of investigations due to the recentness of its introduction.

In light of this trend from coarse to highly localized quantization, whereas aggressive quantization of progressively smaller blocks is being pursued, herein we examine the dynamics of quantization errors across a range of block sizes, uncover a potential pitfall emerging in existing formats and exacerbated by smaller blocks, and propose an efficient hardware design solution to avoid it.

Our contributions are threefold:

- Discovery and analysis of a quantization anomaly in microscaling formats, whereas *decreasing* block size paradoxically *increases* quantization error.

- A theoretical framework that decouples sources of quantization errors and explains their interaction with LLM tensor statistics, allowing us to pinpoint the origin of the anomalous behavior to the quantization of the microscaling scales which hinders the representation of low magnitude blocks. The framework applicability extends beyond the case of FP4 elements and FP8 unsigned E4M3 scales.

- A hardware-friendly design proposals to support FP8 unsigned E5M3 scales to effectively mitigate these issues at minimal hardware cost.

## 2 BACKGROUND

### 2.1 MICROSCALING FORMATS

Microscaling, as defined by the Open Compute Project (OCP, Rouhani et al. (2023a)), refers to a group of numbers sharing a scale and represented as elements with lower precision. The original OCP specification requires elements in either FP4, FP6, INT8, or FP8 format, block size 32, and scales as E8M0, i.e., a biased Power-of-Two (PoT) scale, covering a very wide dynamic range, $2^{-127}$ to $2^{128}$, with limited precision. Since its conception, multiple commercial vendors have added native hardware support for MXFP8 and MXFP4 formats (NVIDIA (2025a); AMD (2025)). In addition, NVIDIA GPUs with Compute Capability 10.0+ support NVFP4, which refers to 16 FP4 elements sharing an FP8 unsigned E4M3 (UE4M3) scale (4 exponent bits, 3 mantissa bits). The selection of a floating point scale in lieu of PoT is motivated by accuracy considerations (Lee et al. (2024); NVIDIA (2025b)), making NVFP4 a strong contender as quantization format of choice for microscaling FP4.

In a typical formulation, microscaling quantization is applied to weights and activations by partitioning each tensor in blocks of size $N$. For each block $j$ with elements $x_i^{(j)}$, a scaling factor $s^{(j)}$ is derived as $s^{(j)} = \mathbb{Q}_{\text{scale}} \left( x_{\max}^{(j)}/C \right)$, where $x_{\max}^{(j)} = \max_{i=1,\dots,N} \left| x_i^{(j)} \right|$, $\mathbb{Q}_{\text{scale}}$ defines the scales quantization, and $C$ is a constant. Each element is normalized by its block scale and mapped into the nearest discrete level as $q_i^{(j)} = \mathbb{Q}_{\text{elem}} \left( x_i^{(j)}/s^{(j)} \right)$, where $\mathbb{Q}_{\text{elem}}$ defines the elements quantization mapping. Values can be reconstructed (i.e., dequantized) by rescaling: $\hat{x}_i^{(j)} = s^{(j)} \cdot q_i^{(j)}$.

### 2.2 RECENT WORK ON MICROSCALING

A Power-of-Two scale simplifies the arithmetic complexity of hardware implementations, but significantly degrades accuracy when applied to FP4 elements (Rouhani et al. (2023b)). Recent works have addressed this issue by proposing techniques that can largely be classified as enhancing the

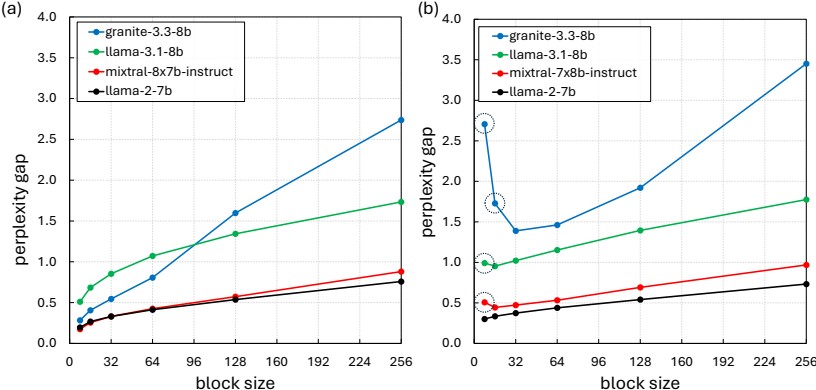

Figure 1: (a) FP4 microscaling quantization using BF16 scales. (b) Impact of FP8 UE4M3 scales. Anomalous data points showcasing perplexity inversion have been circled.

element range for better outliers representation, or improving the scale precision. Lee et al. (2024) use asymmetric FP8 E5M2 scales to mitigate the impact of outliers in 32-element blocks. Chen et al. (2025) propose BitMoD, where the redundant zero representation in the elements is exploited for an additional quantization level. BlockDialect (Jang & Tambe (2025)) uses a 16-way codebook where the two largest elements in a block can be set to 16 configurable pairs. Lo et al. (2024) present Nanoscale floating point, where both the scale format and the element format are adapted based on the distribution of the elements in a block. Collectively, these approaches demonstrate a broader trend toward hardware-efficient quantization, where mixed-format representations, block-level adaptability, and fine-grained floating-point encoding jointly push the limits of low-bit inference while preserving model quality.

## 3 MICROSCALING WITH FINER GRANULARITY

### 3.1 EXPECTED ERROR DEPENDENCY ON BLOCK SIZE

We begin our investigation by monitoring the difference in perplexity (herein, *perplexity gap*) between models quantized with FP4 microscaling formats and their 16-bit precision baseline, as a function of block size. Fig. 1(a) shows the perplexity gap upon FP4 quantization when the scaling factors of each block are kept as BF16, thus *not quantized*. Notably, all models follow a similar trend: as block size is reduced from 256 to 8, the perplexity gap decreases monotonically. This is the generally expected behavior: reducing block size allows for a finer-grain representation of the quantized elements, hence lower quantization error.

To strengthen this intuition, consider for example the quantization of a block $j$ of $N$ elements $x_i$: a single scale $s_0^{(j)}$ is derived from the block maximum $x_{\max}^{(j)}$ (as per Sec. 2.1). If the elements in this block are instead quantized into $B$ separate blocks of size $L = N/B$, each sub-block is assigned a scale $s_b \leq s_0^{(j)}$, for $b = 1, \ldots, B$. Therefore, every element belonging to a sub-block where $s_b < s_0^{(j)}$ is quantized using a scale which, unlike $s_0^{(j)}$, by definition does not exceed the maximum of the sub-block. The smaller scale would typically, although not strictly, enable a better representation of the sub-block elements, resulting in lower error.

The behavior showcased in Fig. 1(a) underpins the motivation to drive per-block quantization towards ever smaller blocks. However, reduced block size comes at a cost, associated with the increased number of quantization scales, which limits the gains in memory storage and data transfer bandwidth provided by the compressed representation. Specifically, memory requirements for a format with $N$ 4-bit elements per block and 16-bit scales, are $1/2 + 2/N$ bytes and every halving of block size increases storage by a factor of $4/(N + 4)$. Moreover, for a scale format with $M$ bits of mantissa (including the implied 1), the complexity of multiplication in a microscaling multiply-and-accumulate operation grows by $M^2 * K$ where $K$ is the width of the partial sum. Therefore,

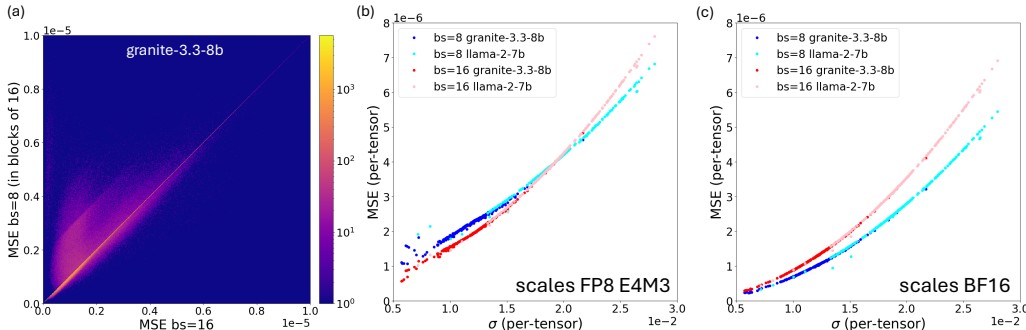

Figure 2: (a) Per-block MSE of the first Query weight tensor, block size (bs) 8 vs 16. (b) Per-tensor MSE vs standard deviation $\sigma$ of each weight tensor of granite-3.3-8b and llama-2-7b. Tensors quantized as FP4 using FP8 UE4M3 scales, with bs 8 or 16. (c) MSE vs $\sigma$ using BF16 scales instead.

irrespective of the block size, the arithmetic complexity of handling 16-bit scale (BF16 or FP16) increases the hardware complexity, compared to an 8-bit scale. Due to these limitations, 8-bit scales have been the de-facto standard since the introduction of FP4 microscaling formats. Scales quantization as FP8 UE4M3, however, brings about unexpected behaviors hinting at a fundamental limit to the block size reduction trend.

## 3.2 ANOMALOUS ERROR DEPENDENCY ON BLOCK SIZE

Figure 1(b) shows the relation between perplexity gap and microscaling block size using FP8 E4M3 scales. Going from large to small values of block size, a monotonic decrease in perplexity gap is initially observed, as in Fig. 1(a), in line with the general expectation outlined in the previous Section. Surprisingly, Fig. 1(b) shows that **a further decrease in block size can lead to an increased perplexity gap**. We refer to this behavior as *perplexity inversion*, or simply inversion. Perplexity inversion is model dependent: granite-3.3-8b shows a clear upswing at block size 16; llama-3.1-8b and mixtral-8x7b-instruct present it at block size 8; on the other hand, llama-2-7b has a strictly monotonic dependence, with no inversion down to block size 8. Inversion (or lack thereof) in additional model architectures is presented in Appendix A.

Although perplexity evaluates performance at the model level, the appearance of perplexity inversion may suggest an increase in the quantization error of each individual weight tensor. To verify this hypothesis, we quantize the same tensor twice with different block sizes, and compute the Mean Squared Error (MSE) *in terms of the larger block* to enable a direct block-to-block comparison. The expectation is that blocks quantized using finer granularity would predominantly show lower MSE. Fig. 2(a) shows the 2D density plots for one weight tensor of granite-3.3-8b, quantized with FP4 elements and FP8 UE4M3 scales using block size 8 and 16. Data points on the diagonal correspond to blocks having the same MSE in both conditions. Strikingly, a large number of data points, about 25% in Fig. 2(a), sit *above the diagonal*, indication that larger quantization error at smaller block size is a frequent occurrence. As the magnitude of the weights never exceeds 1.0 in granite-3.3-8b, this behavior cannot be driven by large outliers truncation. The pattern seen in Fig. 2(a) is consistent across weight tensors and models, as shown in Appendix B. These results provide experimental validation to the counterintuitive hypothesis that quantization error per-block can often increase as block size decreases.

To understand the root causes for the error increase, and why it is differently expressed by different models, in Figs. 2(b,c) we look at the MSE of each weight tensor, computed on a per-tensor basis under a given quantization scheme, as a function of the standard deviation $\sigma$ of the same tensor, pre-quantization. In Fig. 2(b), we quantize with FP4 elements and FP8 UE4M3 scales all weights tensors of granite-3.3-8b and llama-2-7b. The plot shows a distinct dependence of MSE on $\sigma$, in common between the two models. In Appendix C, we demonstrate that this behavior is maintained across a variety of models, from attention-based LLM, to State-Space Models (SSM), to hybrid SSM. Hence, a range of weight distributions, potentially with different tails and outliers incidence,

express the same dependence. We emphasize that a relation between MSE and $\sigma$ is not necessarily surprising: a larger $\sigma$ is representative of a distribution with larger weight magnitudes, and given that the maximum block value determines the normalization factor, it is conceivable that non-maximum elements could incur higher error for a large maximum. In Sec. 4.3 we explore this relation further.

Irrespective of a general dependence, a striking feature emerges from Fig. 2(b) when comparing block size 8 to 16: **the two curves overlap, with a crossover at** $\sigma \approx 2 \cdot 10^{-2}$. Under this empirical threshold, the MSE of weights quantized with block size 8 is larger than their block size 16 counterpart. Granite-3.3-8b, which showed pronounced inversion in Fig. 1(b), is characterized by having most weights below this threshold. On the contrary, llama-2-7b, which showed no inversion in Fig. 1(b), has a large fraction of weights above $\sigma \approx 2 \cdot 10^{-2}$. The crucial role played by the quantized scales in this phenomenon is confirmed in Fig. 2(c), which shows the same curves for non-quantized scales. A monotonic dependence of MSE vs $\sigma$ is still observed, but the error associated with block size 16 quantization is consistently larger than the error using block size 8. These observations provide a first set of insights helping us understand the model-dependent perplexity inversion shown in Fig. 1(b) and guiding towards a solution: models presenting narrow weight distributions quantized using microscaling formats are prone to larger quantization errors when a finer granularity is employed. This behavior is driven by the quantization of the scaling factors.

# 4 THEORETICAL FRAMEWORK

## 4.1 MSE OF IDEAL DISTRIBUTIONS

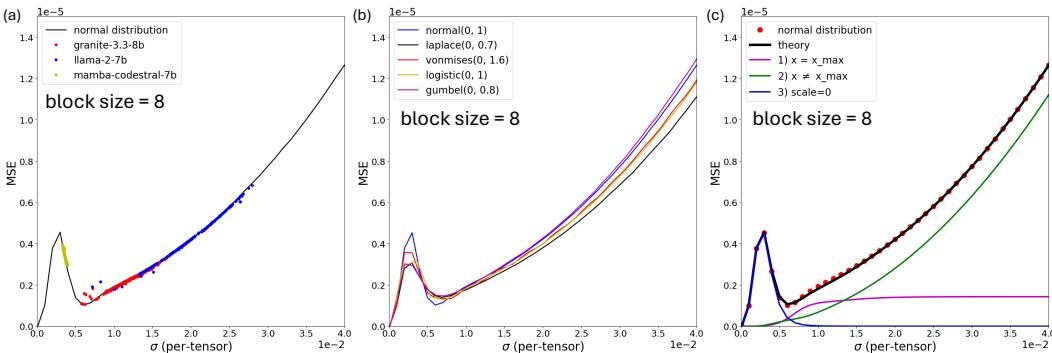

Figure 3: (a) MSE - $\sigma$ dependency for weights of 3 pre-trained models (dots) against a Normal distribution (black line). (b) Behavior of several ideal distributions. (c) Normal distribution compared to theoretical results, and decomposition of 3 contributions to the theoretical error.

To elucidate the mechanisms by which scales quantization drives up the error at smaller block sizes, we investigate the phenomenon from first principles. As a first step towards setting up a theoretical framework, we establish what distribution is appropriate to model. To this extent, we recreate the MSE vs $\sigma$ plot using tensors randomly drawn from various ideal distributions. Tensors are quantized using FP4 elements and FP8 UE4M3 scales. Fig. 3(a) demonstrates an excellent agreement between experimental data from real models and the curve associated with a Normal distribution with mean $\mu = 0$ and variable $\sigma$. At the lowest end of the range of $\sigma$, previously unexplored, for $\sigma < 0.5 \cdot 10^{-2}$ a new feature emerges: as $\sigma$ decreases, the MSE increases dramatically, before dropping again towards zero. To verify this behavior arises in pretrained models as well, we also plot the distribution of weights of mamba-codestral-7b in Fig. 3(a) (yellow dots), which is especially narrow compared to other models. These data also agree with the results from the Normal distribution.

Fig. 3(b) shows the results across multiple ideal distributions. We sweep $\sigma$ by drawing elements with a given distribution and scaling them in magnitude using a range of constants. The parameters determining the shape of each distribution were chosen arbitrarily, aiming at spanning a similar range of $\sigma$ given the same range of scaling factors applied to the drawn tensors. The shape of each distribution (prior scaling) is presented in Appendix D, showing a variety of tail behaviors. Fig. 3(b) highlights that all distributions show qualitatively similar trends, but some deviation from

the Normal distribution curve is also present. Hence, although a Normal distribution appears to be a good approximation of the behavior of pretrained model weights, significant deviations can results in some scattering around the main trend line. The pattern is consistent across block sizes, as shown in Appendix D. Given these observations, we will use a Normal distribution to theoretically model the main trends of the MSE vs $\sigma$.

## 4.2 MICROSCALING FP4 WITH NON-QUANTIZED SCALES

We begin by analyzing the case of microscaling quantization using *non-quantized scaling factors*. We examine this simplified scenario to introduce key concepts and methodologies that will serve as foundation to the more complex case of FP8 UE4M3 scales. We stress that although herein we are focusing on FP4 E2M1 quantization, the framework is readily adaptable to reproduce the behavior of other quantization formats, both floating point and integer. We present key details of this process below, and provide the full derivations in Appendix E.

Consider a random variable $X$ that follows a Normal distribution with mean $\mu = 0$ and standard deviation $\sigma$. We draw $N$ elements $x_i$ from $X$ and define $x_{\max} = \max_{i=1}^{N} |x_i|$. We introduce a second variable $Y$, representing a scaled version of $X$, where the scaling factor $s$ depends on $x_{\max}$: $Y = X/s = 6X/x_{\max}$. The factor 6 is the maximum representable value in the FP4 E2M1 format.

The Probability Density Function (PDF) of $Y$ conditioned on $x_{\max}$ is:

$$f(y \mid x_{\max}) = \frac{\alpha \cdot \phi(\alpha y)}{2\Phi(6\alpha) - 1} \quad \text{for } y \in [-6, 6] \tag{1}$$

with $\phi$ being the PDF of the standard Normal distribution ($\mu = 0$, $\sigma = 1$), $\Phi$ its Cumulative Density Function (CDF), and $\alpha = x_{\max}/(6\sigma) = s/\sigma$.

To derive the MSE conditioned on $x_{\max}$, we consider each FP4 E2M1 quantization level $q_j$ with Voronoi boundaries $[a_j, b_j]$, and compute the conditional MSE of $Y$ for the bin $j$:

$$\text{MSE}_{Y,j}(q_j \mid x_{\max}) = \int_{a_j}^{b_j} (y - q_j)^2 \cdot f(y \mid x_{\max}) \, dy \tag{2}$$

To obtain the MSE as used in Fig. 2(b) and Fig. 3(a,b), we need to express $\text{MSE}_{Y,j}$ with respect to the random variable $Z = s \cdot \mathbb{Q}_{\text{FP4}}(Y)$, which described the elements distribution after discretization $\mathbb{Q}_{\text{FP4}}$ and dequantization. This error, still conditioned on $x_{\max}$ is:

$$\text{MSE}_{Z,j}(q_j \mid x_{\max}) = \frac{\sigma^2}{2\Phi(6\alpha) - 1} \frac{N - 1}{N} \int_{v_j(\alpha)}^{w_j(\alpha)} (u - q_j\alpha)^2 \cdot \phi(u) \, du \tag{3}$$

with $u = \alpha y$, $w_j = a_j\alpha$, and $v_j = b_j\alpha$. In this equation, all terms in $\alpha$ contain the $x_{\max}$ conditionality. To remove the conditioning and obtain the total $\text{MSE}_Z$, we compute the expected value with respect to $x_{\max}$:

$$\text{MSE}_Z = \mathbb{E}_{x_{\max}}\left[\sum_j \text{MSE}_{Z,j}(q_j \mid x_{\max})\right] = \int_0^\infty \sum_j \text{MSE}_{Z,j}(q_j \mid x_{\max}) \cdot f_{x_{\max}}(x) \, dx \tag{4}$$

where $f_{x_{\max}}$ is the PDF of $x_{\max}$, derived as the derivative of the CDF of $N$ i.i.d. variables drawn from a half-Normal distribution:

$$f_{x_{\max}}(\theta) = \frac{2N}{\sigma}\left[2\Phi\left(\frac{\theta}{\sigma}\right) - 1\right]^{N-1} \phi\left(\frac{\theta}{\sigma}\right) \tag{5}$$

We combine eq. 3-5, and integrate numerically over $x_{\max}$ while varying $\sigma$, to obtain an MSE vs $\sigma$ theoretically-derived from first principles, when scales are not quantized. As shown in Fig. 10 (Appendix E), **the agreement with the experimental results from an ideal distribution, and hence with the experimental data from pretrained models, is extremely good** ($\chi^2 \approx 2 \cdot 10^{-9}$).

## 4.3 MICROSCALING FP4 WITH FP8 UE4M3 SCALES

We now introduce into the framework the quantization of the scales as FP8 UE4M3. Full derivation is provided in Appendix F. This condition breaks several previous assumptions. First, the scales

discretization requires us to compute all the contributions to the error of each possible scale value, weighted by the respective probability mass. Second, it is no longer the case that $x_i = x_{\max}$ has zero error. Third, we will have to separately account for the edge case where all the values within a block are rounded to zero.

When $x_i \neq x_{max}$ and $s \neq 0$, the error on the dequantized variable $Z$ can be expressed as:

$$\mathrm{MSE}_{Z,x_i \neq x_{\max}} = \sum_k p_k^{\mathrm{FP8}} \cdot \mathrm{MSE}_{Z,k}(s_k) = \sum_k \int_{a_k}^{b_k} f_S(s)\, ds \cdot \sum_j \mathrm{MSE}_{Z,k,j}(q_j \mid s_k) \quad (6)$$

Here, the index $i$ refers to $N$ elements of a block, $j$ to the quantization levels of the elements, and $k$ to the quantization levels of the scales. $p_k^{\mathrm{FP8}}$ is the probability mass of a scale quantization bin, $a_k$ and $b_k$ its Voronoi boundaries, $f_S(s)$ the PDF of the random variable $S$ associated to the scales. The error of a quantized element $\mathrm{MSE}_{Z,k,j}$ has a similar expression as eq. 3, but conditioned on the scale $s_k$ instead of $x_{\max}$.

When $x_i = x_{max}$ and $s \neq 0$, the error on a single element at a given scale $s_k$ can be computed directly as:

$$\mathrm{Err}_{x_i=x_{\max}}(x, s_k) = \left( \mathbb{Q}_{\mathrm{FP4}}\left( \frac{x}{s_k} \right) \cdot s_k - x \right)^2 \quad (7)$$

and incorporated in the computation of the MSE by summing over all possible $k$ quantization levels of the non-zero scales:

$$\mathrm{MSE}_{Z,x_i=x_{\max}} = \frac{1}{N} \sum_k \int_{6a}^{6b} \mathrm{Err}_{x_i=x_{\max}}(x, s_k) \cdot f_{x_{\max}}(x)\, dx \quad (8)$$

The third contribution to the MSE relates to the lowest FP8 scale bin, associated to $s = 0$, for which eqs. 6 and 8 are ill-defined. In a round-to-nearest process this is the interval $[0, \frac{s_{\min}}{2}]$, where $s_{\min}$ is the lowest non-zero representable FP8 value (subnormal S-0000-001$_b = 2^{-9}$). If the maximum of $N$ values of $X$ generates a scale within this bin, all values in the block are rounded to zero. This error is then:

$$\mathrm{MSE}_{Z,s=0} = P(s=0) \cdot \mathbb{E}\left[ X^2 \mid s=0 \right] = P\left( \frac{x_{\max}}{6} < \frac{s_{\min}}{2} \right) \cdot \mathbb{E}\left[ X^2 \mid |X| < 3s_{\min} \right] \quad (9)$$

The total MSE is the sum of these 3 separate contributions:

$$\mathrm{MSE}_Z = \mathrm{MSE}_{Z,x_i \neq x_{\max}} + \mathrm{MSE}_{Z,x_i=x_{\max}} + \mathrm{MSE}_{Z,s=0} \quad (10)$$

and can be integrated numerically, as in the non-quantized scenario.

Fig. 3(c) compares the estimates for MSE obtained from the theoretical framework (thick black line) against the experimental data derived from a Normal distribution (red dots), under FP4 microscaling quantization with FP8 UE4M3 scales. **The curves are found to be in perfect agreement** ($\chi^2 \approx 4 \cdot 10^{-8}$) **with the experimental data derived from a Normal distribution** (as per Sec. 4.1), across the full interval of $\sigma$, providing strong validation of the theoretical model. Fig. 11 in Appendix F further supports this conclusion by demonstrating that the agreement with the experimental data carries across block sizes and leads to the emergence of crossovers between curves. Fig. 13 in Appendix G demonstrates that the applicability of our theoretical formulation extends across data types, as the modeling of INT4 microscaling quantization also shows very close agreement ($\chi^2 \approx 1.3 \cdot 10^{-6}$) with the experimental data. Similarly, modeling of microscaling FP4 with FP6 scales provides MSE estimates that are consistent with corresponding perplexity measurements (see Appendix H). Hence, in the context of new data format exploration, this framework can play a role in analyzing the impact on the quantization error of scaling down precision, to sub-4-bit element formats, sub-8-bit scales, and smaller block sizes.

Fig. 3(c) also plots the three separate contributions to MSE along with the total MSE. We find that for sufficiently large $\sigma$, the error for all elements $x_i \neq x_{\max}$ dominates the total MSE. The main dependence of MSE on $\sigma$, first highlighted in Fig. 2(b), is therefore due to the superlinear dependence of $\mathrm{MSE}_{Z,j}$ when $x_i \neq x_{\max}$, the quantization error of each FP4 bin at a given scale $s_k$, on $\sigma$, as seen in eq. 3 and 6. However, in narrower distributions the error on the representation of

the maximum value of each block ($x_i = x_{\max}$), which is zero if scales are not quantized, assumes a larger relative importance, and can even dominate the total error. As expected, the smaller the block size, the more the weight of this factor (see Fig. 12 in Appendix F). Finally, at the lowest end of the range of $\sigma$, the error is entirely dominated by the zero rounding of all block elements. **The relative importance of these three separate factors provides direct explanation of the dependence of MSE on block size** (additional commentary in Appendix F.4).

## 5 ERROR MITIGATION STRATEGIES

As demonstrated in Sec. 4, the width of a weight distribution determines the relative importance of different contributions to the total MSE. In particular, narrow distributions appear to be unfavorable, with a large MSE compared to the magnitude of the elements. This behavior would materialize a large relative error and be detrimental to model accuracy. Similar considerations equally apply to activation distributions. Several solutions can be proposed to tackle this problem.

### 5.1 PER-TENSOR SCALING

A recent implementation of FP4 microscaling on GPU (NVIDIA (2025a)) applies a *per-tensor scaling factor* to weight and activations, prior to the FP4 microscaling quantization step with FP8 UE4M3 scales. Such scaling extends the range of narrowly distributed tensors $T$ using the combined maximum representable value of FP4 E2M1 and FP8 UE4M3:

$$s_T = \frac{\max(\text{E2M1}) \cdot \max(\text{UE4M3})}{\max(\text{abs}(T))} \tag{11}$$

The matmul output is then scaled back replacing the denominator of $s_T$ with an estimate of the full precision matmul output, to account for the scaling of both the weight and activation tensors. As shown in Figs. 4(b,c) and Table 1 (as well as Appendix I), this strategy can be effective in improving perplexity and accuracy across a variety of models. However, it also rises two potential concerns. First, per-tensor scaling is susceptible to outliers: a single element of large magnitude hinders the effective scaling of the whole distribution. In this context, when exploring microscaling FP4 applied to attention, in an effort to limit degradation, Zhang et al. (2025) have already empirically resorted to pre-matmul *per-channel* scaling. Second, while the scaling factors associated to weights can generally be precomputed, scaling of activations necessitates to either perform an on-the-fly absmax operation, increasing the computational cost of running the model, or to use pre-calibrated estimates, which may not reflect the actual distribution of the tensor being quantized and introduce errors in the computation. For these reasons, it would be desirable to develop a hardware-friendly strategy to mitigate error while not resorting to the use of a global scale.

### 5.2 FP8 UE5M3

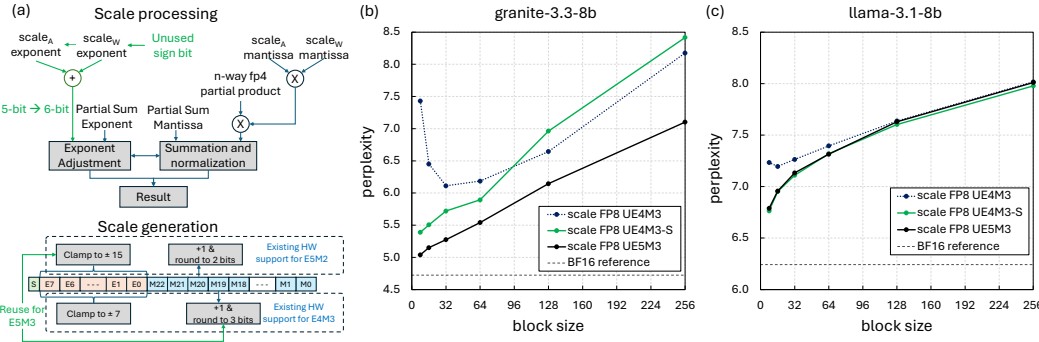

Figure 4: (a) Details of UE5M3 hardware implementation. (b,c) Perplexity vs block size using microscaling FP4 with FP8 UE5M3 scales.

Table 1: Accuracy under the proposed FP4 microscaling quantization schemes, at block size 8. Notation: UE4M3-S: per-tensor scaling + UE4M3; Hsw = HellaSwag; Wng = Winogrande.

| Model | Format | Wiki ↓ | PIQA ↑ | Hsw ↑ | Wng ↑ | GSM8K ↑ | MMLU ↑ |
|---|---|---|---|---|---|---|---|
| granite-3.3-8b | BF16 | 4.72 | 80.41 | 61.49 | 72.38 | 62.47 | 60.55 |
| | UE4M3 | 7.43 | 76.50 | 55.98 | 67.88 | 32.37 | 48.82 |
| | UE4M3-S | 5.39 | 78.84 | 58.86 | 71.27 | 44.88 | 55.23 |
| | UE5M3 (ours) | 5.04 | 79.98 | 60.26 | 73.01 | 56.17 | 57.51 |
| llama-3.1-8b | BF16 | 6.24 | 79.87 | 60.05 | 73.48 | 50.49 | 63.28 |
| | UE4M3 | 7.23 | 78.29 | 57.72 | 72.06 | 32.30 | 56.18 |
| | UE4M3-S | 6.76 | 78.84 | 58.71 | 72.61 | 43.21 | 60.99 |
| | UE5M3 (ours) | 6.79 | 78.84 | 58.94 | 72.14 | 42.15 | 60.97 |
| nemotron-nano-9b-v2 | BF16 | 8.08 | 80.30 | 58.22 | 73.24 | 79.61 | 73.86 |
| | UE4M3 | 8.92 | 79.32 | 57.57 | 70.56 | 72.71 | 71.12 |
| | UE4M3-S | 8.42 | 79.54 | 57.62 | 72.53 | 78.92 | 72.03 |
| | UE5M3 (ours) | 8.39 | 80.03 | 57.57 | 71.19 | 77.71 | 72.29 |
| bamba-9b-v2 | BF16 | 6.21 | 80.96 | 62.18 | 73.95 | 42.15 | 64.96 |
| | UE4M3 | 21.25 | 76.44 | 45.90 | 57.85 | 2.65 | 36.36 |
| | UE4M3-S | 6.53 | 80.41 | 61.64 | 71.90 | 40.41 | 63.80 |
| | UE5M3 (ours) | 6.53 | 80.52 | 61.51 | 72.61 | 39.42 | 64.11 |

As mentioned in Sec. 2.1, existing hardware formats utilize *unsigned* FP8 scales (UE4M3). This choice allows the major hardware vendors to leverage support for existing 8-bit FP8 formats, but leaves one scale bit unused. As an alternative to per-tensor scaling, we propose to repurpose this bit as an exponent and extend the FP8 scale range. The new scale format, unsigned E5M3 (UE5M3) offers greatly increased dynamic range compared to UE4M3, while maintaining its precision. In the context of narrow distributions, this allows a better representation of the scale associated with blocks containing exclusively small magnitude elements: the minimum non-zero representable absolute value drops from $2^{-9}$ for UE4M3 to $2^{-17}$ for UE5M3. Naturally, the extended range would also provide a better representation of large outliers in any given block. An alternative repurposing for the unused bit leading to an UE4M4 format is explored in Appendix J, but is less hardware friendly and found to be less robust than UE5M3.

A UE5M3 scale requires hardware modifications in two key areas: (1) within each microscaling instruction, where scales are fused with elements, and (2) in the quantization operation, where activation outputs are quantized per block to generate scales and elements for the subsequent operation.

**Scale Processing**: Figure 4(a) illustrates the typical floating-point processing steps in AI processors. The UE5M3 format retains the mantissa processing logic of the scale unchanged, while extending the exponent logic by one additional bit. Since mantissa processing primarily determines hardware complexity, this modification introduces only a negligible increase in hardware cost.

**Scale Generation**: AI processors commonly support both E4M3 and E5M2 formats for standard FP8 quantization, and may reuse the same hardware features for generating scales in MX-FP4 quantization. To support E4M3 and E5M2, existing hardware must include logic to cast an 8-bit exponent (from a FP32 value) to either a 4-bit or 5-bit exponent. Similarly, it must be capable of rounding a 23-bit mantissa to either 3-bit or 2-bit precision. Consequently, UE5M3 scale generation can be achieved with minimal hardware changes.

**Hardware Design**: to demonstrate the feasibility of the UE5M3 format at hardware level, the solution was incorporated into the design of a systolic array processing engine with a microarchitecture similar to that described in Agrawal et al. (2021). Additional details and overhead estimates are provided in Appendix K.

Fig. 4(b,c) and Table 1 show that applying FP8 UE5M3 scale to FP4 weights and activations *without per-tensor scaling* achieves better or comparable performance as using FP8 UE4M3 scales with per-tensor scaling (UE4M3-S). We emphasize we computed the per-tensor scaling *dynamically* for UE4M3-S, thus the results reflect the best accuracy that can be achieved with this format. Fig. 16 and Table 3 in Appendix I show the UE5M3 results hold across block sizes and for a variety of models, from attention-based LLM, to SSM, to hybrid SSM-attention models.

## 6 CONCLUSIONS

Microscaling formats are poised to become in the near future the format of choice for model quantization. Proper understanding of the sources of error associated with these formats is critical to avoid potentially costly pitfalls at training and inference. In this paper, we analyzed the unexpected trends that quantization errors can present when quantizing narrow distributions with microscaling formats. We demonstrated that such anomalous behavior is a direct consequence of the quantization of the scaling factors, and introduced a theoretical framework to identify the separate contributions driving this effect. Theoretical results are in remarkable agreement with experimental observations. On the basis of this understanding, we have proposed a hardware-friendly implementations, FP4 microscaling with FP8-UE5M3 scales, that effectively and efficiently mitigates these errors.

## 7 REPRODUCIBILITY STATEMENT

We share our UE5M3 implementation and our theoretical model at `https://github.com/i clr2016codeshare/microscaling`. Full derivation of the theoretical model is provided in Appendices E and F.

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

## A    PERPLEXITY GAP ACROSS VARIOUS MODELS AND LOW BLOCK SIZE

In our experiments to measure perplexity, we quantize both weights and activations of all linear layers except the last one (model head), with the selected microscaling quantization format. Attention matmul are not quantized. Perplexity is computed based on a next-token prediction task on the Wikitext2 dataset (Merity et al. (2016)), using the test split, with samples of sequence length 2048 tokens. We use perplexity as a proxy metric for accuracy, as a starting point to highlight potential issues associated with the quantization process. We provide a comprehensive analysis of accuracy across multiple benchmarks in Sec. 5.2 and in Appendix I.

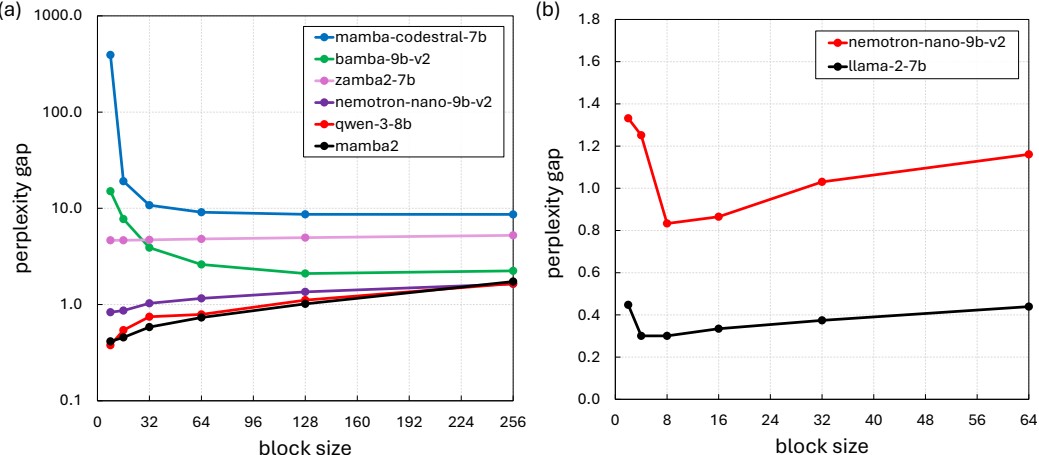

Figure 5: (a) Perplexity gap vs. block size for FP4 with FP8-UE4M3 scales quantization. Plotted with a logarithmic y-axis to account for the significant perplexity inversion observed in some models. (b) Even when perplexity inversion is not present down to block size 8 (as llama-2-7b in Fig. 1(b)), it can still emerge at even smaller block sizes (2, 4).

# B   PER-BLOCK MSE: BLOCK SIZE 8 VS 16

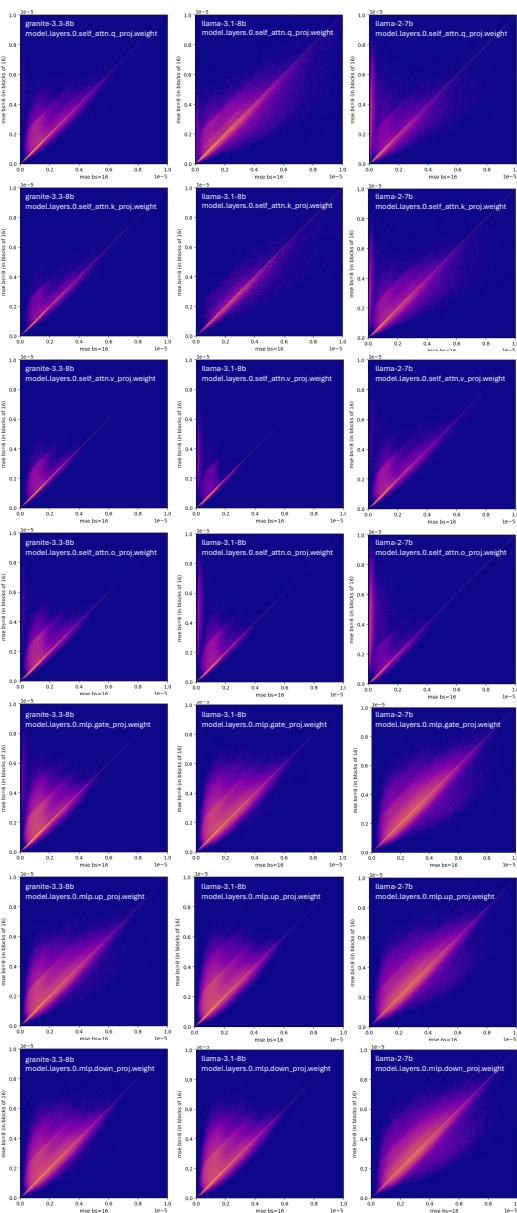

Figure 6: Comparison of per-block MSE for block size 8 vs 16 microscaling quantization, using FP4 elements with FP8-UE4M3 scales. As shown by the prevalence of data points above the diagonal, errors using the smaller block size routinely exceed errors with larger block size, across different weight tensors and models. This pattern holds even for models not showing perplexity inversion at block size 8.

# C  MSE VS $\sigma$ OF MODELS AND BLOCK SIZES

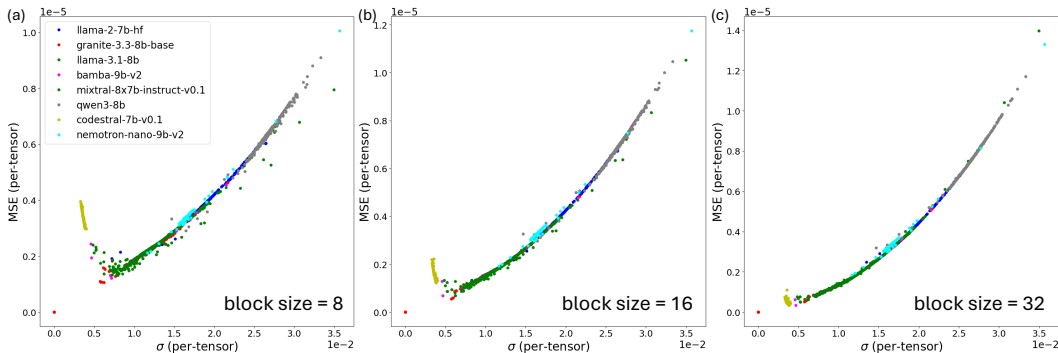

Figure 7: Per-tensor MSE vs standard deviation $\sigma$ of each weight tensor of various LLMs, highlighting a common dependence across models. Very narrow weights may appear to be deviating from the predominant trend, but this behavior is also present in ideal distributions and captured by our theoretical framework (see Fig. 3).

# D  MSE vs $\sigma$ across ideal distributions

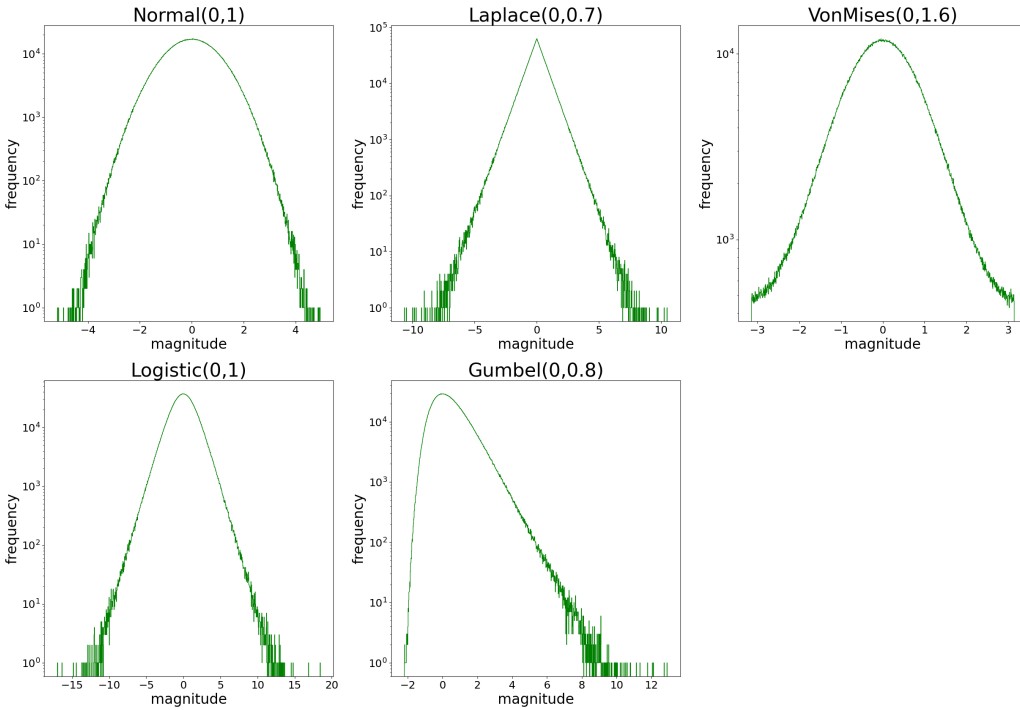

Figure 8: Shape of the ideal distributions used to compute the MSE vs $\sigma$ curve in Fig. 3(b), showcasing different tails and asymmetries. We stress that the selected parameters are arbitrary and were only chosen such that the corresponding $\sigma$ for each distribution type would cover a similar interval, upon application of the same vector of scaling factors.

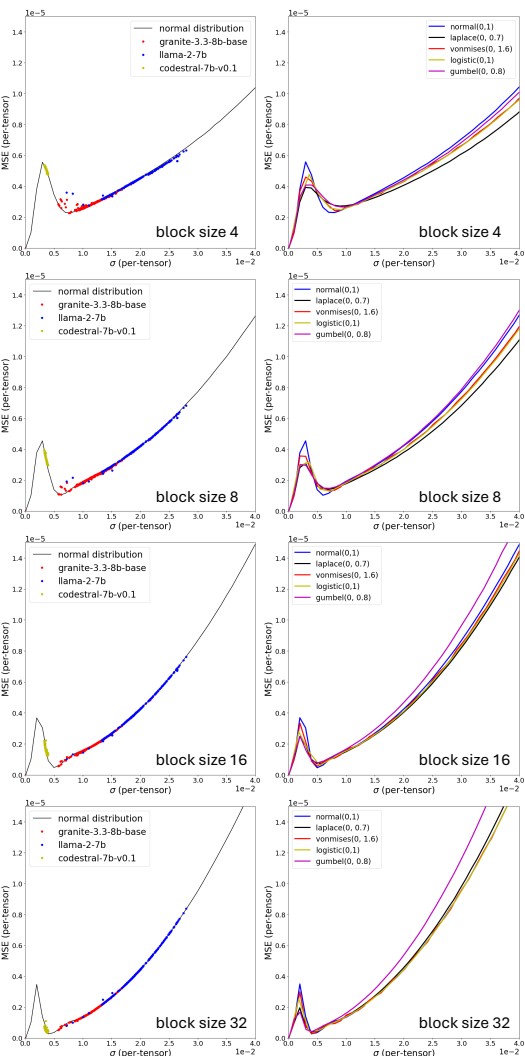

Figure 9: (left column) MSE vs $\sigma$ comparison of experimental data from 3 models and a Normal distribution of mean 0 and variable $\sigma$. Across multiple block sizes, the data point from the pre-trained models are in excellent agreement with their Normal counterpart. Notice the sharp increase towards very narrow distributions is observed at any block size. (right column) MSE vs $\sigma$ from experimental data drawn from different distributions.

# E   THEORETICAL FRAMEWORK: NON-QUANTIZED SCALES

In this appendix, we present the full derivation of the theoretical framework in the case microscaling quantization with scales not quantized (i.e., infinite precision is assumed), and a comparison of theoretically-derived MSE-$\sigma$ curves against experimental data from a Normal distribution (Fig. 10).

As mentioned in Sec. 4.2, we model the drawing of $N$ elements $x_i$ from a random variable $X$ with probability density $P(X)$. $P(X)$ is a Normal distribution of mean 0 and standard deviation $\sigma$: $X \sim \mathcal{N}(0, \sigma^2)$. We use $X$ to mimic weight distribution. We quantize these elements using a symmetric block-wise FP4 E2M1 quantization scheme, with non-quantized scale defined as

$$s = \frac{x_{\max}}{m} \tag{12}$$

with $x_{\max} = \max_{i=0}^{N-1} |x_i|$. The factor $m$ is the maximum representable value for the data format used for element quantization. For example, $m = 6.0$ for FP4 E2M1. The scale $s$ is used to derive the scaled elements prior quantization, $y_i$, as

$$y_i = \frac{x_i}{s} = \frac{mx_i}{x_{\max}} \tag{13}$$

Accordingly, the scaled elements $y_i$ to be quantized are drawn from a random variable $Y$ with *scaled and truncated* gaussian distribution:

$$Y = \frac{mX}{x_{\max}} \tag{14}$$

The distribution of $Y$ is truncated to $[-m, m]$ because $x_i \leq x_{\max} \ \forall i$. Post-quantization, the de-quantized elements $z_i$ belong to the random variable $Z$:

$$Z = \frac{x_{\max}}{m} \mathbb{Q}(Y) = \frac{x_{\max}}{m} \mathbb{Q}\left(\frac{mX}{x_{\max}}\right) = s \cdot \mathbb{Q}\left(\frac{X}{s}\right) \tag{15}$$

with $\mathbb{Q}$ being the mapping function to the quantized levels defined by the selected element format (e.g., FP4 E2M1).

The first step towards deriving the MSE of $Z$ is to compute the PDF of $Y$ conditioned to $x_{\max}$. As the MSE associated with $x_i = x_{\max}$ is zero, we will only derive an expression for the PDF associated with the $N - 1$ elements satisfying $x_i \neq x_{\max}$. For $x_i = x_{\max}$, the PDF is in the form of two Dirac functions located at $\pm x_{\max}$ but as it does not contribute to the MSE, we will leave it out of the derivation.

We define $\tilde{f}_Y(y)$ as the distribution of $Y$ prior normalization. We derive $\tilde{f}_Y(y)$ by applying the change of variable formula for the transformation of a continuous random variable, from $X$ to $Y = g(X)$. With $y = g(x) = \frac{mx}{x_{\max}} \Rightarrow g^{-1}(y) = \frac{x_{\max}y}{m}$, we get:

$$
\begin{aligned}
\tilde{f}_{Y, x_i \neq x_{\max}}(y \mid x_{\max}) &= f_X\left(g^{-1}(y)\right) \cdot \left|\frac{d}{dy} g^{-1}(y)\right| \\
&= f_X\left(\frac{x_{\max}y}{m}\right) \cdot \left|\frac{x_{\max}}{m}\right| \\
&= \frac{1}{\sqrt{2\pi}\sigma} \exp\left(-\frac{1}{2\sigma^2} \cdot \left(\frac{x_{\max}y}{m}\right)^2\right) \cdot \frac{x_{\max}}{m} \\
&= \frac{x_{\max}}{m\sigma\sqrt{2\pi}} \exp\left(-\frac{1}{2} \cdot \left(\frac{x_{\max}y}{m\sigma}\right)^2\right) \\
&= \frac{\alpha}{\sqrt{2\pi}} \exp\left(-\frac{(y\alpha)^2}{2}\right) \\
&= \alpha \cdot \phi(\alpha y) \quad \text{for } y \in [-m, m]
\end{aligned}
\tag{16}
$$

with

$$\alpha = \frac{x_{\max}}{m\sigma} = \frac{s}{\sigma} \tag{17}$$

and $\phi$ the Probability Density Function (PDF) of the *standard* ($\mu = 0$, $\sigma = 1$) Normal distribution:

$$\phi(y) = \frac{1}{\sqrt{2\pi}}\exp\left(-y^2/2\right) \tag{18}$$

To obtain $f_Y(y)$, the PDF of Y, we normalize $\tilde{f}_Y(y)$ considering that the probability density is zero outside $[-m, m]$:

$$f_{Y,x_i \neq x_{\max}}(y \mid x_{\max}) = \frac{\tilde{f}_Y(y \mid x_{\max})}{\int_{-m}^{m} \alpha\phi(\alpha y)dy} = \frac{\tilde{f}_Y(y \mid x_{\max})}{\int_{-m\alpha}^{m\alpha} \phi(u)du} = \frac{\alpha \cdot \phi(\alpha y)}{2\Phi(m\alpha) - 1} \tag{19}$$

Here, $\Phi$ is the Cumulative Density Function (CDF) of the standard Normal distribution.

Next, we compute the conditional expected square error on $Z$, the distribution after dequantization. We focus on the error on $Z$, instead of $\mathbb{Q}(Y)$, for a direct comparison with the experimental data in Fig. 2(b,c) and Fig. 3(a-c). This error is the sum of the errors associated to each quantization bin $\mathrm{MSE}_{Y,j}(q_j \mid x_{\max})$. Each quantization bin comprises one of the quantization levels $q_j$ (with $j = 1, \ldots, N_Q$ and $N_Q$ the number of quantization levels of the format used to quantize the elements) and is defined by the Voronoi boundaries $[a_j, b_j]$. The bin error is:

$$\begin{aligned}
\mathrm{MSE}_{Y,j}(q_j \mid x_{\max}) &= \frac{N-1}{N}\int_{a_j}^{b_j}(y - q_j(y))^2 f_{Y\mid x_{\max}}(y)\,dy \\
&= \frac{N-1}{N}\int_{a_j}^{b_j}(y - q_j(y))^2 \frac{\alpha \cdot \phi(\alpha y)}{2\Phi(m\alpha) - 1}
\end{aligned} \tag{20}$$

where the $(N-1)/N$ factor takes into account that only $N-1$ elements $x_i$ contribute to the error.

The square error with respect to $Z$ is related to the square error on $Y$ via the scaling factors $s = x_{\max}/m$ as:

$$X - Z = \frac{x_{\max}}{m}(Y - \mathbb{Q}(Y)) \Rightarrow (X - Z)^2 = \left(\frac{x_{\max}}{m}\right)^2 (Y - \mathbb{Q}(Y))^2 \tag{21}$$

Therefore, the error becomes:

$$\begin{aligned}
\mathrm{MSE}_{Z,j}(q_j \mid x_{\max}) &= s^2 \mathrm{MSE}_{Y,j} \\
&= s^2\int_{a_j}^{b_j}(y - q_j)^2 \cdot f(y \mid x_{\max})\,dy \\
&= (\alpha\sigma)^2 \frac{N-1}{N}\int_{a_j}^{b_j}(y - q_j)^2 \frac{\alpha \cdot \phi(\alpha y)}{2\Phi(m\alpha) - 1}\,dy \\
&= \frac{\sigma^2}{2\Phi(6\alpha) - 1}\frac{N-1}{N}\int_{v_j(\alpha)}^{w_j(\alpha)}(u - q_j\alpha)^2 \cdot \phi(u)\,du
\end{aligned} \tag{22}$$

We remove the conditioning on $x_{\max}$ by computing its expected value:

$$\mathrm{MSE}_Z = \mathbb{E}_{x_{\max}}\left[\sum_j \mathrm{MSE}_{Z,j}(q_j \mid x_{\max})\right] = \int_0^\infty \sum_j \mathrm{MSE}_{Z,j}(q_j \mid x_{\max}) \cdot f_{x_{\max}}(x)\,dx \tag{23}$$

All that is left is to compute $f_{x_{\max}}(x)$. As $x_{\max}$ is computed with an absmax operation, its distribution is one of $N$ i.i.d. variables drawn from the *half-Normal distribution* $\Theta$:

$$\Theta = |X|, \quad \Theta \sim \mathrm{HalfNormal}(\sigma) \tag{24}$$

The half-Normal CDF with $\theta_i$ drawn from $\Theta$ is the known formula:

$$F_{x_{\max}}(\theta) = 2\Phi\left(\frac{\theta}{\sigma}\right) - 1 \tag{25}$$

and since $|\theta_i|$ are i.i.d. the probability of drawing $N$ variables with $P(|X| \leq x_{\max})$ is the product of each individual probability:

$$\mathbb{P}(|X| \leq x_{\max}) = \mathbb{P}(|x_i| \leq x \, \forall i = 1, \ldots, N) = [\mathbb{P}(|x_i| \leq x)]^N \tag{26}$$

This is expressed by the product of $N$ CDF:

$$F_{x_{\max}}(\theta) = \left[ 2\Phi\left(\frac{\theta}{\sigma}\right) - 1 \right]^N \tag{27}$$

The PDF is obtained by differentiating this CDF:

$$
\begin{aligned}
f_{x_{\max}}(\theta) = \frac{d}{d\theta} F_{x_{\max}}(\theta) &= N \left[ 2\Phi\left(\frac{\theta}{\sigma}\right) - 1 \right]^{N-1} \frac{d}{d\theta}\left( 2\Phi\left(\frac{\theta}{\sigma}\right) - 1 \right) \\
&= 2N \left[ 2\Phi\left(\frac{\theta}{\sigma}\right) - 1 \right]^{N-1} \frac{d}{du}\Phi(u)\frac{du}{d\theta} \\
&= 2N \left[ 2\Phi\left(\frac{\theta}{\sigma}\right) - 1 \right]^{N-1} \phi\left(\frac{\theta}{\sigma}\right) \frac{1}{\sigma} \\
&= \frac{2N}{\sigma} \left[ 2\Phi\left(\frac{\theta}{\sigma}\right) - 1 \right]^{N-1} \phi\left(\frac{\theta}{\sigma}\right)
\end{aligned} \tag{28}
$$

As the integral from 0 to $\infty$ of $f_{x_{\max}}$ is 1, this is already a PDF, no additional normalization is needed. Using this formulation for $f_{x_{\max}}$, the MSE using non-quantized scales is:

$$\mathrm{MSE}_Z = \int_0^\infty \sum_j \mathrm{MSE}_{Z,j}\left(q_j \mid x_{\max}\right) \cdot \frac{2N}{\sigma} \left[ 2\Phi\left(\frac{\theta}{\sigma}\right) - 1 \right]^{N-1} \phi\left(\frac{\theta}{\sigma}\right) \, dx \tag{29}$$

$$\mathrm{MSE}_Z = \int_0^\infty \mathrm{MSE}_Z\left(u\right) \cdot f_{x_{\max}}(u) \, du \tag{30}$$

$$\mathrm{MSE}_Z = \mathbb{E}_{X_{\max}}\left[ \mathbb{E}\left[ (X - Z)^2 \mid X_{\max} \right] \right] \tag{31}$$

$$\mathrm{MSE}_Z(u) := \mathbb{E}\left[ (X - Z)^2 \mid X_{\max} = u \right] \tag{32}$$

Finally, we compute this error by discretizing the domain of $x_{\max}$ over a series of $\theta$, computing all $\theta$-dependent parameters (implicit in all $\alpha$ dependencies), and integrating numerically. The resulting $\mathrm{MSE}_Z$ can be compared directly with the per-tensor MSE obtained experimentally, from either pre-trained models or ideal distributions.

As shown in Fig. 10, the agreement between theoretical model and experimental data is very remarkable, across a range of $\sigma$ and block sizes $bs$.

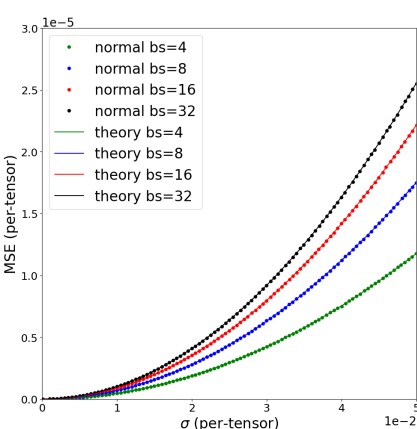

Figure 10: Per-tensor MSE vs standard deviation $\sigma$ curves comparing estimates from the theoretical framework with *non-quantized scales* against corresponding results obtained experimentally from elements drawn from a Normal distribution. The remarkable overlap validates the correctness of the theoretical framework.

# F  THEORETICAL FRAMEWORK: FP8 UE4M3 SCALES

In this appendix, we present the full derivation of the theoretical framework in the case of microscaling quantization with FP8-UE4M3 scales. This derivation naturally builds on the process described in Appendix E and summarized in Sec. 4.2, where we derived a formulation in the case of non-quantized scales.

When scales are FP8, they are additionally quantized with a "casting" operation to FP8:

$$s = \mathbb{Q}_{\text{FP8}} \left( \frac{x_{\max}}{m} \right) \tag{33}$$

As before, $m$ is the maximum representable value of the quantized *elements* ($m = 6.0$ for FP4 E2M1). $\mathbb{Q}_{\text{FP8}}$ is the operator that maps each high-precision scale value to its FP8 counterpart. Different quantization schemes can be modeled by simply replacing the $\mathbb{Q}$ operator and summing over the $k$ quantization levels associated with the new mapping.

When dealing with non-quantized scales, we only considered the $x_i \neq x_{\max}$ case, because the element $x_i = x_{\max}$ was represented exactly, and thus did not contribute to the error. This consideration resulted in an $(N-1)/N$ scaling factor applied to the MSE (see eq. 3). When scales themselves are quantized, we must also account for the $x_i = x_{\max}$ scenario, resulting in non-zero quantization error. This error is to be scaled by $1/N$ instead of $(N-1)/N$, accounting for 1 element out of $N$ block elements being used for scale derivation in this condition.

In addition, we will need to separately account for the case where all elements within a block are rounded to zero. This occurs when $\frac{x_{\max}}{m} < \frac{s_{\min}}{2}$, with $s_{\min}$ being the lowest non-zero representable scale value. This contribution is not included in the previous derivation, where the case $s = 0$ was both ill-defined and contributed no error due to the continuous nature of the non-quantized scales.

Overall, we will be modeling 3 separate sources of error:

- $x_i \neq x_{\max}$ with FP8 scales $s \neq 0$
- $x_i = x_{\max}$ with FP8 scales $s \neq 0$
- FP8 scales $s = 0$

## F.1  $x_i \neq x_{\max}$ WITH SCALES $s \neq 0$

The case of $x_i \neq x_{\max}$ with $s \neq 0$ was covered in the non-quantized scales framework by removing the conditioning over $x_{\max}$ via integration. However, when introducing a discretization of the scales, we will have to sum over each quantized scale level $s_k$, so it becomes more convenient to express all formulas in terms of this parameter. We start by deriving an expression for $f_{S_x}(s)$, the distribution of the random variable $S_X$, representing the continuous scales prior quantization. This can be computed using the dependence of $S_X$ on $x_{\max}$, with a change of variable to the function $f_{x_{\max}}(x)$, the previously-computed distribution of $x_{\max}$. We start from:

$$s = g(x) = \frac{x}{m} \Rightarrow g^{-1}(s) = m \cdot s \Rightarrow \frac{dx}{ds} = m \tag{34}$$

and obtain the expression for the distribution:

$$
\begin{aligned}
f_{S_X}(s) &= f_{x_{\max}}(m \cdot s) \cdot \left| \frac{dx}{ds} \right| \\
&= f_{x_{\max}}(m \cdot s) \cdot m \\
&= m \cdot \frac{2N}{\sigma} \left[ 2\Phi\left( \frac{m \cdot s}{\sigma} \right) - 1 \right]^{N-1} \phi\left( \frac{m \cdot s}{\sigma} \right)
\end{aligned}
\tag{35}
$$

To get the probability mass $p_i^{\text{FP8}}$ of a single scale quantization bin around a value $s_k$, we can integrate within the bin's Voronoi boundaries $a_k, b_k$, assuming round-to-nearest:

$$p_k^{\text{FP8}} = \text{P}(\mathbb{Q}(s) = s_k) = \int_{v_k}^{w_k} f_{S_X}(s) \, ds \tag{36}$$

To remove the conditioning on $s_k$, instead of integrating over the continuous distribution $f_{x_{\max}}$, we sum over the $k$ FP8 bins the product of $p_k^{\text{FP8}}$ and $\text{MSE}_{Z,k}(s_k)$, the latter being the MSE with respect to the random variable $Z$ of the quantized/dequantized elements, conditioned to a scale $s_k$. Deriving $\text{MSE}_{Z,k}(s_k)$ requires knowledge of the PDF of $Y$:

$$f_Y(y \mid s_k) = \frac{\tilde{f}_Y(y \mid s_k)}{\int_{-m}^{m} \alpha\phi(\alpha y)\, dy} = \frac{\alpha_k \cdot \phi(\alpha_k y)}{2\Phi(m\alpha_k) - 1} \tag{37}$$

with $\alpha_k = s_k/\sigma$. Notice that for $s_k = 0$, the normalization factor denominator of $f_Y(y \mid s_k)$ is zero. Hence, we will have to treat the $s_i = 0$ case separately.

The expression for the MSE per-bin is similar to the non-quantized scale scenario, except for the dependence on $k$:

$$\text{MSE}_{Z,k,j}(q_j \mid s_k) = s_k^2 \int_{a_j}^{b_j} (y - q_j)^2 \cdot f(y \mid s_k)\, dy$$

$$= \frac{\sigma^2}{2\Phi(6\alpha_k) - 1} \frac{N - 1}{N} \int_{v_j(\alpha_k)}^{w_j(\alpha_k)} (u - q_j\alpha_k)^2 \cdot \phi(u)\, du \tag{38}$$

where the $(N-1)/N$ factor addresses the condition $x_i \neq x_{\max}$.

Finally, we remove the conditioning on $s_k$ by summing over the scales quantization levels $k$ and the elements quantization levels $j$:

$$\text{MSE}_{Z,x_i \neq x_{\max}} = \sum_k p_k^{\text{FP8}} \cdot \text{MSE}_{Z,k}(s_k)$$

$$= \sum_k \int_{a_k}^{b_k} f_{S_X}(s)\, ds \cdot \sum_j \text{MSE}_{Z,k,j}(q_j \mid s_k) \tag{39}$$

### F.2 $x_i = x_{\max}$ WITH SCALES $s \neq 0$

When $x_i = x_{\max}$ and scales are not quantized, the error on $x_i$ is zero: the scale is derived directly from $x_{\max}$ and one quantization level is aligned to $x_{\max}$ exactly. However, this is no longer the case upon scale quantization, as the scale $\mathbb{Q}_{\text{FP8}}$ introduces quantization error.

We can compute the error conditional on $x_{\max}$ directly, as it pertains a single element $x_i = x_{\max}$, in lieu of having to integrate over the distribution of $y$:

$$\text{Err}_{x_i = x_{\max}}(x, s_k \mid x_{\max}) = \left( \mathbb{Q}_{\text{elem}}\left(\frac{x}{s_k}\right) \cdot s_k - x \right)^2 \tag{40}$$

where $\mathbb{Q}_{\text{elem}}$ is the element quantization mapping (e.g., FP4 E2M1). Notice this computation is not defined for $s_k = 0$, the third source of error to be treated separately.

The corresponding MSE for the scale bin $k$ ($\text{MSE}_{Z,x_i = x_{\max},k}$) is derived by simply scaling $\text{Err}_{x_i = x_{\max}}$ by $1/N$, because the error $\text{Err}_{x_i = x_{\max}}$ only applies to a single element in a block of size $N$.

To obtain the total MSE across all scales, we integrate $\text{MSE}_{Z,x_i = x_{\max},k}$ along with the probability mass $p_k^{\text{FP8}}$ of a scales $s_k$, and sum over all non-zero scales:

$$\text{MSE}_{Z,x_i = x_{\max}} = \sum_k p_k^{\text{FP8}} \cdot \text{MSE}_{Z,x_i = x_{\max},k}$$

$$= \frac{1}{N} \sum_k \int_{ma_k}^{mb_k} \text{Err}_{x_i = x_{\max}}(x, s_k) \cdot f_{x_{\max}}(x)\, dx \tag{41}$$

The interval $[ma_k, mb_k]$ comes from the boundaries $[a_k, b_k]$ of the bin of $s_k$, expressed in terms of $x = m \cdot s_k$.

## F.3 SCALES $s = 0$

The third and last source of MSE error comes from the lowest FP8 scale bin which, assuming a round-to-nearest process, goes from 0 to $\frac{s_{\min}}{2}$, where $s_{\min}$ is the lowest non-zero representable FP8 value ($2^{-9}$ for IEEE FP8 E4M3, which includes subnormals). If the maximum of $N$ values of $X$ generates a scale that falls within this bin, all values in the block are rounded down to zero. This increases the error for ultra-narrow distributions.

The error for zero quantized scale is:

$$\text{MSE}_{s=0} = P(s = 0) \cdot E[X^2 \mid s = 0] \tag{42}$$

where $E[X^2 \mid s = 0]$ is the expected error when the scale is zero, and $P(s = 0)$ is the probability of having zero scale. The probability of zero scale $P(s = 0)$ is derived considering that once a value $x_{\max}$ is drawn, all other $x_i$ satisfy the condition $x_i < x_{\max}$. Hence, $x_i$ do not conform to the original distribution of $X$ with standard deviation $\sigma$ but to a distribution that is truncated in the interval $\left[-\frac{m s_{\min}}{2}, \frac{m s_{\min}}{2}\right] = [-b_{\min}, b_{\min}]$. Therefore:

$$P(s = 0) = P(x_{\max} < b_{\min}) = \left(F_{|X|}(b_{\min})\right)^N \tag{43}$$

where $F_{|X|}(x)$ is the CDF of $|X|$ that we already computed as eq. 25. The expected error with a zero scale is computed using the known expression (which includes a normalization factor at the denominator):

$$E[X^2 \mid s = 0] = E[X^2 \mid |X| < b_{\min}] = \frac{\int_{-b_{\min}}^{b_{\min}} x^2 f_X(x)\, dx}{\int_{-b_{\min}}^{b_{\min}} f_X(x)\, dx} \tag{44}$$

## F.4 TOTAL ERROR

The total MSE when using quantized FP8 scales is the sum of the 3 separate contributions we computed in the previous sections:

$$\text{MSE}_Z = \text{MSE}_{Z, x_i \neq x_{\max}} + \text{MSE}_{Z, x_i = x_{\max}} + \text{MSE}_{Z, s=0} \tag{45}$$

The first two terms contain a block size-dependent scaling factor: $(N-1)/N$ or $1/N$, respectively. Both these terms sum over all $s_i \neq 0$, while $\text{MSE}_{Z, s=0}$ is a single scale bin computation. As in the case of non-quantized scales, we compute $\text{MSE}_Z$ by discretizing the domain of $x_{\max}$ and integrating numerically.

Fig. 11 compares theoretical results (lines) against experimental data derive by quantizing elements from a Normal distribution (dots). The agreement is found to be excellent across a wide range of $\sigma$ and various block sizes, capturing all features of the experimental curves, including the presence and location of crossover points, and steep MSE increase for ultra-narrow distributions.

Fig. 12 expands Fig. 3(c), presenting the individual contribution of the 3 sources of errors for block size 4, 8, 16, and 32. The relative weight to the total error of the different contributions varies. At large block size, the error is largely dominated by $\text{MSE}_{Z, x_i \neq x_{\max}}$. As block size is decreased, $\text{MSE}_{Z, x_i = x_{\max}}$ becomes more prominent, materializing the crossover between curves of different block size. At small block size, $\text{MSE}_{Z, s=0}$ impacts a wider interval of lower-end $\sigma$ and its magnitude increases. Hence, narrow distributions become more and more susceptible to this source of error as block size is decreased.

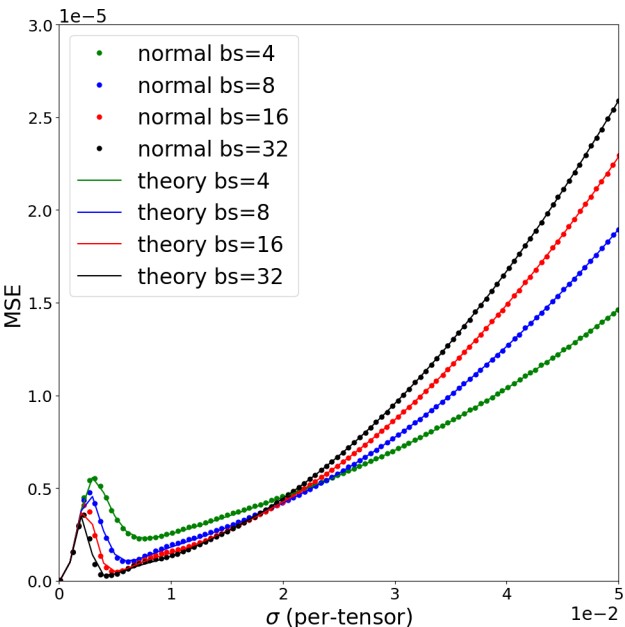

Figure 11: MSE estimates from the theoretical framework for FP4 microscaling quantization with FP8 UE4M3 scales (lines) against corresponding results obtained experimentally from elements drawn from a Normal distribution (dots).

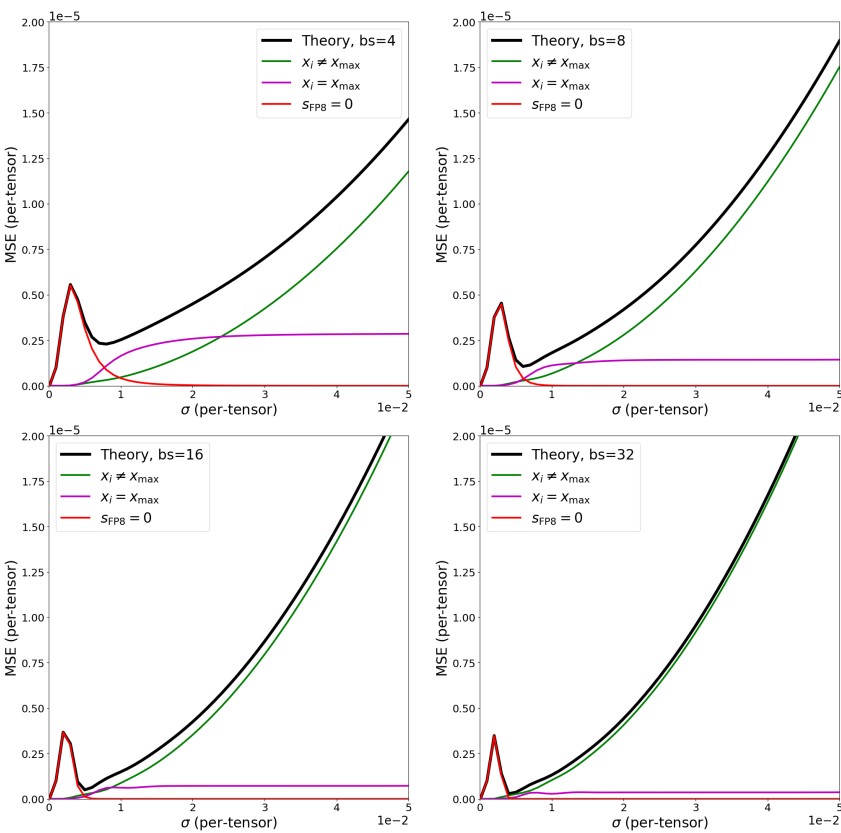

Figure 12: Separate contributions to quantization error for block sizes 4, 8, 16, and 32.

## G    MICROSCALING INT4 QUANTIZATION WITH FP8 UE4M3 SCALES

To provide further validation of the flexibility of our framework in modeling custom quantization strategies, we applied simple modifications to extract the MSE estimates under INT4 element per-block quantization (instead of FP4), with FP8 UE4M3 scales. This change only required an update to the specified list of quantization levels and to the simulation parameters that are dependent on the maximum of the element data format. For asymmetric INT4 quantization, which quantizes in range $[-7, 7]$, the format maximum is 7, instead of 6.0 of FP4. The comparison against experimental data obtained by directly quantizing elements drawn from Normal distribution and computing the corresponding per-tensor MSE is shows in Fig. 13. Similarly to the FP4 case, theoretical results and experimental data are found in excellent agreement ($\chi^2 = 1.3 \cdot 10^{-6}$).

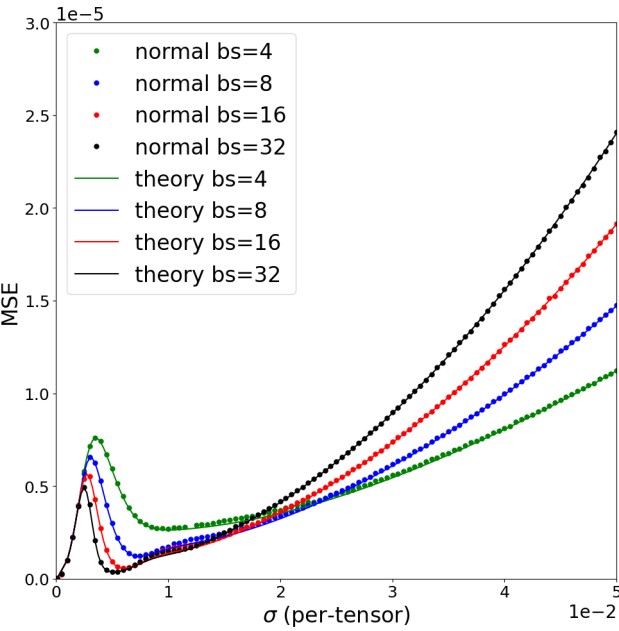

Figure 13: Theoretical MSE estimates against experimental data from Normal distribution, under microscaling INT4 quantization with FP8 UE4M3 scales.

Perplexity evaluation on microscaling INT4 formats across various models (Fig. 14) shows that perplexity inversion is still observed, albeit in a less pronounced manner compared to microscaling FP4. Both UE4M3-S and UE5M3 are effective in mitigating the inversion and improving performance, with UE5M3 achieving comparable or better perplexity than UE4M3-S.

These results are consistent with the analysis of ideal normal distributions (synthetic data) as well as our theoretical framework (Fig. 13) for microscaling INT4: both show that narrow element distributions quantized as INT4 do experience a qualitatively similar effect as FP4, with smaller block size producing larger error than larger block size in narrow distributions. For block size 16 vs. 8, the crossover does occur at a lower standard deviation ($\approx 1.5 \cdot 10^{-2}$) than for FP4 ($\approx 2 \cdot 10^{-2}$), suggesting that considering all tensors within a model, and their spread of standard deviations, we should observe a less pronounced effect, possibly arising at lower block sizes than FP4, at which the increase in error is larger.

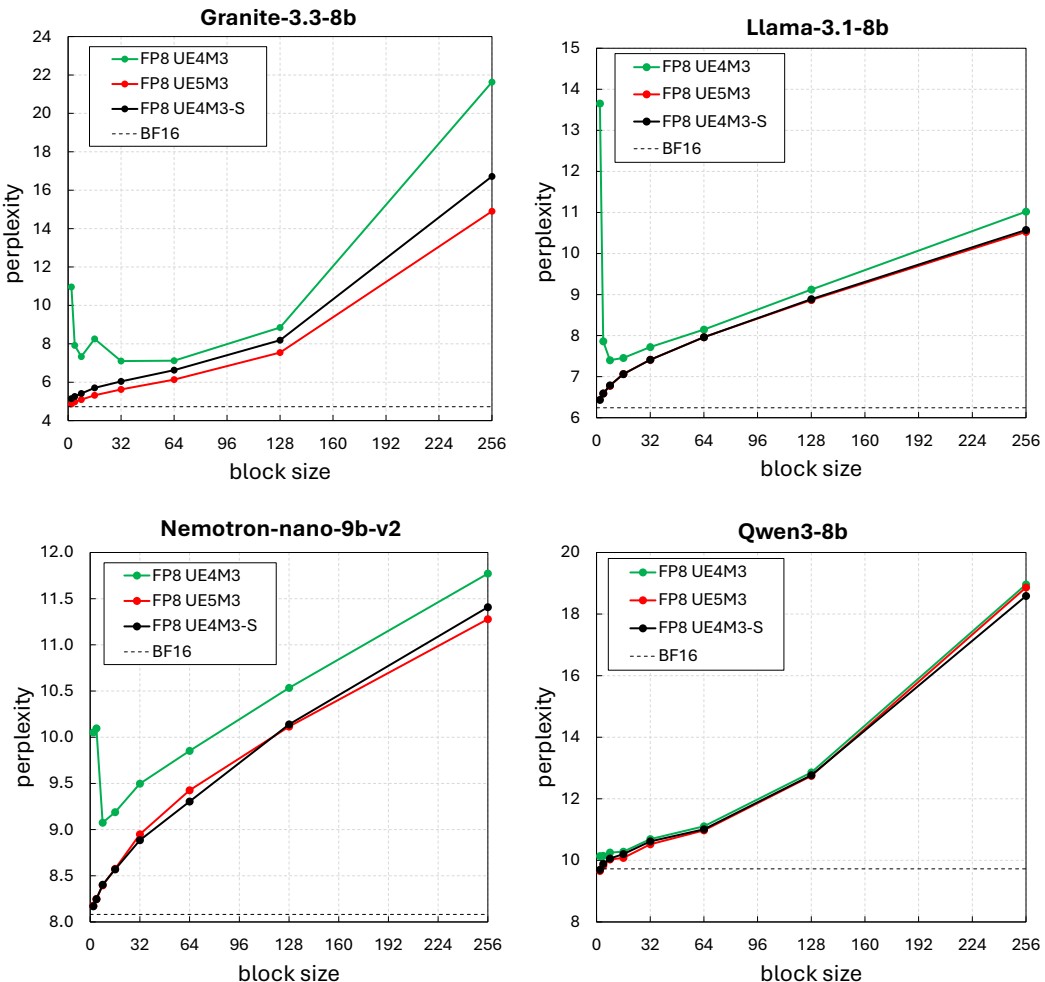

Figure 14: Perplexity evaluation of microscaling INT4 with different scales: UE4M3, UE4M3-S, and UE5M3.

## H  MICROSCALING FP4 QUANTIZATION WITH FP6 SCALES

As further validation of our framework, we have measured the impact of microscaling FP4 with two custom FP6 formats. We stress that no standard format has been recommended for FP6 *scales* at this time, and while OCP (Rouhani et al. (2023a)) prescribes FP6 E3M2 and FP6 E2M3 for elements quantization in MXFP6 formats, based on our findings herein reported we believe supporting the widest dynamic range is of utmost importance. Hence, we tested FP6 UE5M1 and FP6 UE4M2 scales (assuming repurposing of the unused sign bit), in combination with FP4 E2M1 elements, to quantize weights and activations of llama-3.1-8b (BF16 baseline perplexity = 6.242). Consistently with UE4M3-S nomenclature, UE5M1-S and UE4M2-S represent the use of per-tensor scale on both weights and activations.

Table 2: Perplexity (the lower the better) of llama-3.1-8b using microscaling FP4 elements at variable block size and two FP6 formats for scales, UE5M1 and UE4M2, with or without per-tensor scaling. BF16 baseline is 6.242

| Block size | UE5M1 | UE5M1-S | UE4M2 | UE4M2-S |
|---:|---:|---:|---:|---:|
| 2 | 7.217 | 7.240 | 46 795.660 | 6.598 |
| 4 | 7.265 | 7.280 | 343.621 | 6.726 |
| 8 | 7.316 | 7.346 | 19.641 | 6.872 |
| 16 | 7.414 | 7.424 | 11.398 | 7.038 |
| 32 | 7.519 | 7.547 | 9.348 | 7.182 |
| 64 | 7.715 | 7.696 | 9.607 | 7.385 |
| 128 | 7.955 | 7.952 | 10.955 | 7.660 |
| 256 | 8.328 | 8.307 | 9.017 | 8.019 |

Results are entirely consistent with our previous conclusions related to FP8 scales: the wider dynamic range provided by UE5M1 achieves limited degradation in absence of a per-tensor scale. In fact, applying such global scale has negligible effect on perplexity when using this format. However, compared to UE5M3, this format suffers from the loss of 2 bits of precision. On the other hand, UE4M2 performs very poorly without per-tensor scale and we once again we observe a *perplexity inversion* behavior. This can be effectively mitigated by means of a per-tensor scale (UE4M2-S).

With respect to the validation of our theoretical framework, we modeled these two microscaling formats based on FP4 E2M1 elements and FP6 scales, either UE5M1 and UE4M2. MSE estimates are reported in Fig. 15(a,b) across different block sizes. In the case of FP6 UE5M1 scales, no crossover is observed between curves of different block size, consistently with absence of perplexity inversion reported above. By contrast, MSE estimates for FP6 UE4M2 show similar features as FP8 E4M3 (as in Fig. 3c and Fig. 11), but accentuated by the moderate decrease in dynamic range and lower precision brought about by the smaller mantissa. Interestingly, for FP6 UE4M2, crossover occurs at larger $\sigma$ compared to FP8 E4M3 (for example, at $\sigma \approx 3.8 \cdot 10^{-2}$ going from block size 16 to 8, compared to $\sigma \approx 2 \cdot 10^{-2}$ we reported in Sec. 3.2), which suggests even wider distributions, and hence more models, would experience larger error at smaller block sizes. Hence, mitigation becomes even more important as more aggressive quantization formats are deployed.

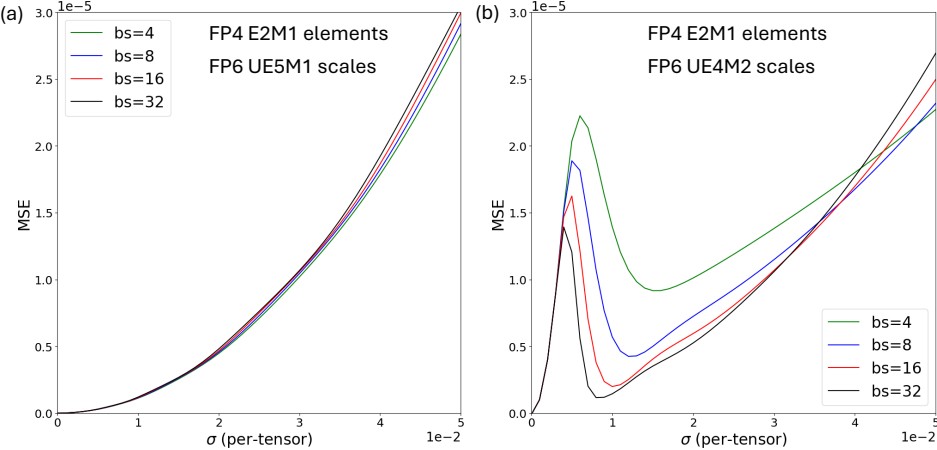

Figure 15: Theoretical MSE estimates for microscaling FP4 quantization with two FP6 scales formats (without per-tensor scaling)

# I ACCURACY USING UE5M3 VS. UE4M3 WITH PER-TENSOR SCALING

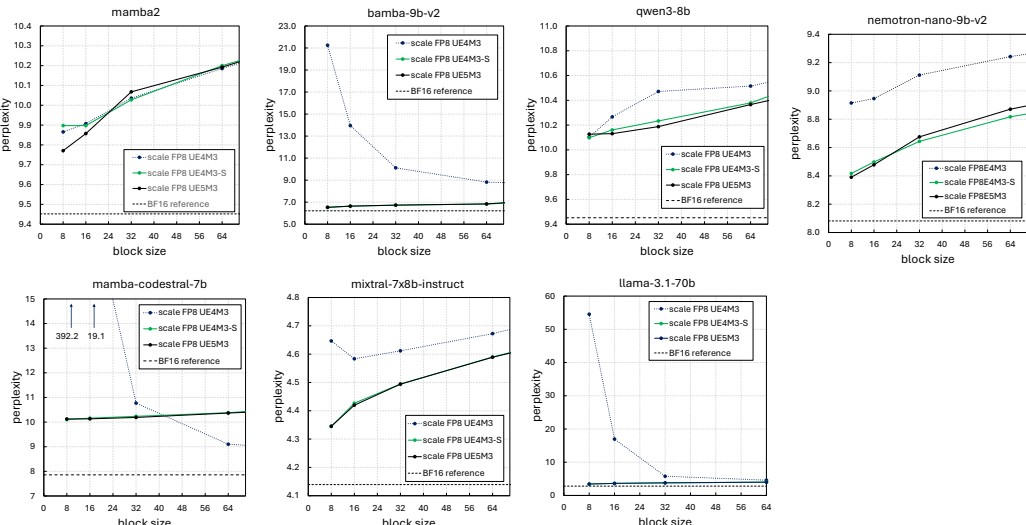

Figure 16: Across various LLMs, UE5M3 scales perplexity is consistently on par with UE4M3 scales using per-tensor scaling, and shows large improvements with respect to UE4M3 scales without per-tensor scaling.

Table 3: Accuracy under the proposed FP4 microscaling quantization schemes, at **block size 16**. Acronyms are the same as in Tab. 1

| Model | Format | Wiki ↓ | PIQA ↑ | Hsw ↑ | Wng ↑ | GSM8K ↑ | MMLU ↑ |
|---|---|---|---|---|---|---|---|
| granite-3.3-8b | BF16 | 4.72 | 80.41 | 61.49 | 72.38 | 62.47 | 60.55 |
| | UE4M3 | 6.45 | 78.50 | 56.98 | 70.71 | 30.17 | 50.67 |
| | UE4M3-S | 5.51 | 77.42 | 58.58 | 72.22 | 36.47 | 54.44 |
| | UE5M3 (ours) | 5.15 | 79.71 | 60.08 | 71.11 | 51.25 | 56.12 |
| llama-3.1-8b | BF16 | 6.24 | 79.87 | 60.05 | 73.48 | 50.49 | 63.28 |
| | UE4M3 | 7.20 | 77.91 | 57.96 | 70.64 | 36.09 | 56.37 |
| | UE4M3-S | 6.95 | 78.29 | 58.73 | 72.69 | 39.27 | 59.16 |
| | UE5M3 (ours) | 6.96 | 78.51 | 58.50 | 71.74 | 38.13 | 58.96 |
| nemotron-nano-9b-v2 | BF16 | 8.08 | 80.30 | 58.22 | 73.24 | 79.61 | 73.86 |
| | UE4M3 | 8.95 | 79.60 | 57.57 | 70.24 | 71.03 | 71.49 |
| | UE4M3-S | 8.50 | 79.60 | 57.46 | 73.80 | 76.50 | 71.78 |
| | UE5M3 (ours) | 8.48 | 79.49 | 57.02 | 73.17 | 74.15 | 72.13 |
| bamba-9b-v2 | BF16 | 6.21 | 80.96 | 62.18 | 73.95 | 42.15 | 64.96 |
| | UE4M3 | 13.95 | 78.13 | 51.72 | 65.59 | 10.39 | 44.85 |
| | UE4M3-S | 6.64 | 79.92 | 61.50 | 73.64 | 38.89 | 63.11 |
| | UE5M3 (ours) | 6.64 | 80.14 | 61.17 | 72.22 | 40.49 | 63.63 |

## J    ALTERNATIVE BIT REPURPOSING: FP8 UE4M4 SCALES

The unused bit of FP8 E4M3 can be alternatively repurposed to extend the mantissa, instead of the exponent. The resulting FP8 UE4M4 format for scales not only benefits from higher precision, but also a moderately extended dynamic range: the lowest representable subnormal element decreases from $2^{-9}$ to $2^{-10}$. Based on our findings, this is expected to lower the error associated to scales quantization. Fig. 17 confirms that UE4M4 is indeed beneficial, but UE5M3 remains the superior solution, more effective and robust across various block sizes. In addition, it is important to remark that, as mentioned in Section 3.1, the complexity of multiplication in hardware scales quadratically with the number of mantissa bits $M$, once again favoring UE5M3 as the most hardware friendly option.

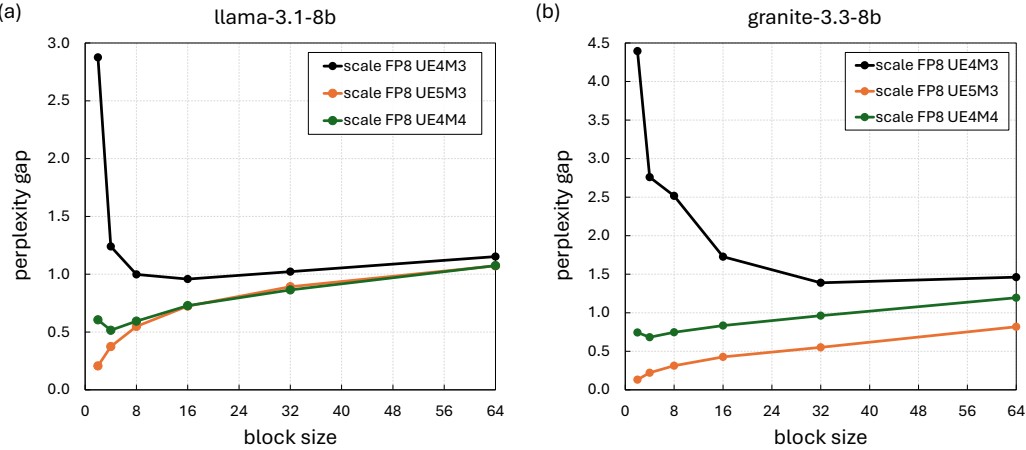

Figure 17: Perplexity gap for alternative scale format FP8 UE4M4.

# K UE5M3 HARDWARE DESIGN

We synthesized a systolic array processing engine (PE) with a microarchitecture similar to that described in Agrawal et al. (2021). The engine has eight Single Instruction Multiple Data (SIMD) lanes, and each lane contains multiple multiply-and-accumulate (MAC) engines, each performing MAC operations on multiple weights and input terms, corresponding to different precisions. Each SIMD lane supports BF16, FP8 (both E4M3 and E5M2), INT8, and microscaling FP4. Two versions of microscaling FP4 were synthesized: one with an E4M3 scale and the other with an E5M3 scale.

Both E4M3 and E5M3 incur the same multiplier cost for processing the sum of FP4 product terms and the product of the scale mantissas. E5M3 requires a 5-bit adder to compute the product scale exponent, compared to a 4-bit adder for E4M3. The resulting product exponent is further subtracted from the 8-bit exponent of the inter-PE partial sum; therefore, the width of the subsequent adders/datapath remains unchanged.

Logic synthesis was performed using a 4 nm process node in a production-grade EDA flow. The area for the E5M3 scale is 0.5% larger than that for the E4M3 scale, which is negligible and does not affect the bounding box area for place-and-route or subsequent SoC floorplanning. Critical path timing increases by 4 picoseconds, which is negligible for setting the SoC frequency. The intuition behind this small area/timing impact is that the effect of the wider adder is diluted by the arithmetic pipelines for other precisions, as well as non-arithmetic logic such as operand staging and the local register file for operand reuse.