# OpenReview forum: "Is Finer Better? The Limits of Microscaling Formats in Large Language Models"
_ICLR.cc/2026/Conference — ICLR 2026 Poster_

### Official Review · Reviewer_UtsX · 2025-10-27

**Soundness:** 3
**Presentation:** 3
**Contribution:** 2
**Rating:** 6
**Confidence:** 4

**Summary:**

This paper investigates the counterintuitive behavior in microscaling quantization where smaller block sizes can lead to higher quantization error. The authors provide both experimental evidence across several LLMs and a theoretical framework explaining the phenomenon, attributing it to scale quantization effects. They further propose a hardware-friendly fix using FP8-UE5M3 scale representation, demonstrating improved performance.

**Strengths:**

The discovery of the “perplexity inversion” phenomenon is novel and well-motivated.

The theoretical modeling is rigorous and matches experimental data convincingly.

The proposed UE5M3 solution is simple, practical, and hardware-friendly.

Writing and figures are clear; experiments cover multiple models and tasks.

**Weaknesses:**

The experiments mainly focus on inference; it would strengthen the paper to evaluate whether the same anomaly occurs during training.

While hardware feasibility is discussed qualitatively, more quantitative data (e.g., area, latency, or energy cost of adding one exponent bit) would clarify the trade-offs.

The proposed format is only tested on sub-10B models. Given the claim of generality, evaluating larger-scale LLMs (e.g., 30B–70B) would enhance credibility.

Some connections to existing FP8 and mixed-precision deployment standards (e.g., NVIDIA MXFP4, OCP spec) could be more explicitly compared.

**Questions:**

How does the anomaly behave with integer quantization (INT4) in practical LLM inference, not only synthetic distributions?

Would dynamic or learned scale clipping alleviate the same issue without new hardware?

---

> ### Author Response · Authors · 2025-11-20
> **Answer to reviewer UtsX**
>
> We thank the reviewer for the pointed comments and suggestions, which served as the basis to improve our publication. Please find below our detailed answer the various points that were raised in the review.
>
> **1. Does the same anomaly occur during training?**
>
> We agree with the reviewer that the training case is valuable to examine. While our experiments target inference, the anomaly we identify is not specific to inference: it arises from the quantization of block-wise scales themselves. As shown in Sec. 3, this manifests as an increased MSE at the tensor level (Fig. 2a) for layers whose distributions have small variance (Fig. 2b). Such layers appear in both forward and backward passes during training.
>
> Therefore, whenever training operates in a regime where many tensors have small magnitude, scale quantization with UE4M3 is expected to be impacted by the anomalous quantization error we identified. This implies that the anomaly may not only be present in training but could affect optimization, potentially requiring the optimizer to compensate by shifting weights toward larger magnitudes, which may lead to sub-optimal convergence or altered minima.
>
> Both mitigation strategies we evaluate for inference, UE4M3-S and UE5M3, address the same root cause of the anomaly, and therefore are applicable during training as well. In particular, UE5M3 avoids the need for global scaling, at both training and inference time.
>
> A comprehensive empirical study of the training dynamics under UE5M3 is part of our planned future work.
>
> **2. Provide quantitative data on hardware feasibility**
>
> We thank the reviewer for raising this important question, which gives us the opportunity to expand on the hardware aspects of our exploration. Albeit details of the full hardware implementation are beyond the scope of this paper, we are glad to share the following estimates that allow us to emphasize that UE5M3 is a realistic hardware solution. We have added these details in Appendix K.
>
> We synthesized a systolic array processing engine (PE) with a microarchitecture similar to that described in [1]. The engine has eight Single Instruction Multiple Data (SIMD) lanes, and each lane contains multiple multiply-and-accumulate (MAC) engines, each performing MAC operations on multiple weights and input terms, corresponding to different precisions. Each SIMD lane supports BF16, FP8 (both E4M3 and E5M2), INT8, and microscaling FP4. Two versions of microscaling FP4 were synthesized: one with an E4M3 scale and the other with an E5M3 scale.
>
> Both E4M3 and E5M3 incur the same multiplier cost for processing the sum of FP4 product terms and the product of the scale mantissas. E5M3 requires a 5-bit adder to compute the product scale exponent, compared to a 4-bit adder for E4M3. The resulting product exponent is further subtracted from the 8-bit exponent of the inter-PE partial sum; therefore, the width of the subsequent adders/datapath remains unchanged.
>
> Logic synthesis was performed using a 4 nm process node in a production-grade EDA flow. The area for the E5M3 scale is 0.5 % larger than that for the E4M3 scale, which is negligible and does not affect the bounding box area for place-and-route or subsequent SoC floorplanning. Critical path timing increases by 4 picoseconds, which is negligible for setting the SoC frequency. The intuition behind this small area/timing impact is that the effect of the wider adder is diluted by the arithmetic pipelines for other precisions, as well as non-arithmetic logic such as operand staging and the local register file for operand reuse.
>
> [1] Agrawal, A. et al. "9.1 A 7nm 4-core AI chip with 25.6 TFLOPS hybrid FP8 training, 102.4 TOPS INT4 inference and workload-aware throttling", 2021 IEEE International Solid-State Circuits Conference (ISSCC), vol. 64, 144-146, 2021

---

> > ### Author Response · Authors · 2025-11-20
> > **Answer to reviewer UtsX (2)**
> >
> > **3. Evaluation of LLMs beyond 10B parameters**
> >
> > Given that the anomalous behavior is a consequence of the interplay between tensor elements magnitude and block-wise scale quantization, we are confident such effect would arise in larger LLMs in the presence of narrowly distributed weights or activations, when using a naive UE4M3 scale.
> >
> > As a preliminary demonstration in this direction, we followed the reviewer's advice and expanded our investigation by evaluating perplexity on wikitext of llama-3.1-70b (using the same setup as we used for other perplexity evaluations in our paper; BF16 baseline = 2.809) and expanded Fig. 16 in Appendix I to include these new data.
> >
> >     |         Llama-3.1-70b perplexity         |
> >     | Block Size |  UE4M3  | UE4M3-S |  UE5M3  |
> >     |------------+---------+---------+---------|
> >     |         8  | 54.550  |  3.442  |  3.446  |
> >     |        16  | 16.973  |  3.628  |  3.641  |
> >     |        32  |  5.779  |  3.778  |  3.796  |
> >     |        64  |  4.604  |  4.035  |  3.983  |
> >     |       128  |  9.470  |  4.763  |  5.257  |
> >     |       256  |  7.193  |  6.749  |  5.821  |
> >
> > These results are consistent with our expectations and confirm that perplexity inversion is present in this model quantized with microscaling FP4 and FP8 UE4M3 scales. Both UE4M3-S and UE5M3 scales dramatically improve perplexity and restore the expected monotonic perplexity trend. Our proposed format does so without the need of additional per-tensor scaling operations.
> >
> > **4. Clarify connections with FP8 and other microscaling formats**
> >
> > We thank the reviewer for this pointer. We have reworded the description of standard microscaling formats in section 2.1 and fixed an incorrect statement about MXFP4 scales. Even though studies presenting a direct comparisons between MXFP4 and NVFP4 are scarce at this time, likely due to the novelty of the format and the limited hardware support, some preliminary indications (Fig. 4 in [1]) suggest that the combination of finer granularity and higher scale precision allows NVFP4 to achieve better performance than MXFP4. Hence, our focus in exploring formats relying on scales with narrower dynamic range, but higher precision, than MXFP4.
> >
> > Regarding comparison to a well-established, non-microscaling formats such as FP8-E4M3 (signed), these typically results in limited degradation compared to its 16-bit counterpart. Even under the computationally-inexpensive Direct Quantization scheme that we employ throughout our paper, which avoids backpropagation entirely, accuracy is on par with BF16. As an example, to mirror our experiments with microscaling FP4, we applied FP8-E4M3 per-channel quantization to weights and activations (excluding attention matmul and SSM states, wherever present) across various models. Results are comparable to the BF16 baselines we used as references, to the point it would be hard to clearly visualize these data in Fig. 1a,b or Fig. 16.
> >
> >     |                     |    wikitext perplexity   |
> >     | Model               |  BF16   | FP8 W&A per-ch |
> >     |---------------------+---------+----------------|
> >     | llama-3.1-8b        |  6.242  |  6.306         |
> >     | granite-3.3-8b      |  4.724  |  4.774         |
> >     | qwen3-8b            |  9.725  |  9.793         |
> >     | codestral-7b        |  7.858  |  7.877         |
> >     | bamba-9b-v2         |  6.214  |  7.128         |
> >     | nemotron-nano-9b-v2 |  8.082  |  8.066         |
> >     | mamba2              |  9.452  |  9.512         |
> >     | mixtral-7x8-instr   |  4.139  |  4.183         |
> >     | llama-3.1-70b       |  2.809  |  2.869         |
> >
> > [1] https://developer.nvidia.com/blog/fine-tuning-gpt-oss-for-accuracy-and-performance-with-quantization-aware-training/

---

> > > ### Author Response · Authors · 2025-11-20
> > > **Answer to reviewer UtsX (3)**
> > >
> > > **5. Inference evaluation of microscaling INT4**
> > >
> > > We agree with the reviewer that it would be valuable to expand our study of microscaling INT4 (Appendix G) to further validate our observations and the predictions of our theoretical framework. To this purpose, we have now added Fig. 14 to Appendix G, which supplements the theoretical analysis with measurements of perplexity across INT4 element-quantization formats using FP8 scales. To collect these data, we used the same methodology we employed across the paper for FP4 quantization, and compared results for UE4M3, UE4M3-S, and UE5M3 scales.
> > >
> > > Importantly, perplexity inversion is still observed across these models, albeit in a less pronounced manner. Both UE4M3-S and UE5M3 are effective in mitigating the inversion and improving performance, with UE5M3 achieving comparable or better perplexity than UE4M3-S.
> > >
> > > These results are consistent with the analysis of ideal normal distributions (synthetic data) as well as our theoretical framework (Fig. 13 in Appendix G): both show that narrow element distributions quantized as INT4 do experience a qualitatively similar effect as FP4, with smaller block size producing larger error than larger block size in narrow distributions. The crossover does occur at a lower standard deviation ($\approx 1.5 \cdot 10^{-2}$) than for FP4 ($\approx 2 \cdot 10^{-2}$), suggesting that considering all tensors within a model, and their spread of standard deviations, we should observe a less pronounced effect, possibly arising at lower block sizes than FP4, at which the increase in error is larger. This is indeed what Fig. 13 and the newly added Fig. 14 show.
> > >
> > > **6. Would dynamic or learned scale clipping alleviate the same issue without new hardware?**
> > >
> > > Dynamic or learned clipping are reasonable directions to explore, and we thank the reviewer for highlighting them. The methods we evaluate already rely on dynamic, absmax-based scale generation, both 1) within each block, to determine the block-wise microscaling scale (for UE4M3, UE4M3-S, or UE5M3), and 2) at the tensor level, for the global scaling strategy (UE4M3-S).
> > >
> > > In contrast to calibration-based absmax approaches, dynamic strategies are calibration-free attempts to ensure full coverage of the range of the tensor (for the global scale) and each block (for the block-wise scales). However, as shown in Sec. 3 and 4, even such strategies cannot compensate for the representational floor imposed by UE4M3’s minimum nonzero scale. Hence, dynamic strategies do not eliminate the errors associated with narrow distributions. Alternative approaches (such UE4M3-S or UE5M3) are needed.
> > >
> > > In principle, more sophisticated dynamic methods, such as alternative clipping rules, percentile clipping, or learned clip values (as suggested by the reviewer), may help, as long as they aren't limited to determine the optimal boundaries for quantization, but also push tensor distributions into a more quantizable range. Simply altering the quantization boundaries (clips) or scale cannot eliminate the quantization error associated with small-magnitude blocks, which already utilize, when using absmax, the smallest representable scaling factor permitted by a given data format. We note, however, that such retraining methods require additional compute budget, necessitate the use of calibration datasets and, to the best of our knowledge, they have not yet been studied thus far in the context of microscaling quantization. We view learned clip optimization for microscaling formats as an interesting avenue of investigation, one that however may not necessarily bypass the representation limit of UE4M3. UE5M3 resolves this limit at the hardware level with minimal additional complexity, allowing microscaling to function robustly even for very narrow weight and activation distributions.

---

### Official Review · Reviewer_bRVj · 2025-10-31

**Soundness:** 2
**Presentation:** 2
**Contribution:** 2
**Rating:** 2
**Confidence:** 3

**Summary:**

This paper uncovers a surprise in LLM quantization: making quantization blocks too small ("finer") can paradoxically hurt performance. The authors trace this "perplexity inversion" anomaly to the standard FP8 UE4M3 format used for the per-block scales, which fails to accurately represent tensors with very small values (narrow distributions). They provide a rigorous theoretical model to prove this and propose a simple, hardware-friendly fix: FP8 UE5M3, a new scale format that uses a spare bit to add a 5th exponent bit. This new format solves the anomaly and achieves high accuracy without requiring expensive per-tensor scaling operations.

**Strengths:**

Identifies the counter-intuitive "finer is worse" quantization anomaly.

Develops a mathematical framework that perfectly explains the why behind the anomaly, which is a significant step beyond just observing it.

**Weaknesses:**

The theory is heavily based on weight distributions (modeled as Normal), with less focus on how the anomaly impacts different and often asymmetric activation distributions.

The claim of "minimal" hardware cost for UE5M3 is asserted but not analyzed in-depth (e.g., no area or latency estimates).

**Questions:**

How does this anomaly, and the UE5M3 fix, perform with the different, often-asymmetric distributions of activations?

Did you investigate adding a mantissa bit (i.e., UE4M4) for precision instead of an exponent bit (UE5M3) for range?

Table 1 shows that for several models (e.g., llama-3.1-8b, bamba-9b-v2), your UE5M3 proposal achieves nearly identical performance to the UE4M3-S (with scaling), not a clear gain. Given this, what is the primary motivation for a hardware change when a software mitigation performs comparably?

---

> ### Author Response · Authors · 2025-11-20
> **Answer to reviewer bRVj (1)**
>
> We thank the reviewer for taking the time to review our submission and for providing constructive criticisms, which gives us the chance to clarify key aspects of our investigation. Below, we address each concern individually, providing additional context and data in support of our study.
>
> **1. Presence of anomaly and impact of FP8 UE5M3 in the context of asymmetric activation distributions**
>
> We acknowledge our theoretical framework is built on the assumption of normally distributed elements, and the reviewer is correct in pointing out that the distribution of activations may deviate from this assumption. However, there are strong reasons to believe a similar effect as described in the paper, whereas smaller block size leads to higher quantization error, would still arise in the presence of asymmetrically distributed elements. In Fig. 3b the comparison of various distributions, including a highly asymmetric Gumbel distribution (whose shape is shown in Appendix D, Fig. 8), shows _qualitatively_ similar behavior at block size 8. Fig. 9 in Appendix D, expands these results across block sizes. Given the similar trends between different distributions, we selected to model a Normal distribution for our framework as it would allow us to obtain close form analytical results, while capturing comparable features of the MSE-sigma plots.
>
> Furthermore, by comparing the plots in Fig. 9, we notice that a crossover from a region where large block size has larger error than small block size, to the opposite behavior which triggers perplexity inversion, would still take place for the asymmetric distribution, albeit at different $\sigma$ than for Normal distributions. For example, comparing block size 16 to 8, the crossover for this particular Gumbel distribution is found at $\sigma \approx 1.5 \cdot 10^{-2}$, instead of the $\approx 2 \cdot 10^{-2}$ crossover point for Normal distribution that we mention in Section 3.2. Overall, these data support the claim that _narrowly-distributed_ asymmetric distributions can still be negatively impacted by scale quantization, and would benefit from the larger dynamic range provided by UE5M3.
>
> If a specific model were to fail due to a deviation from Normal distribution, we argue that microscaling INT4 (in combination with FP8 UE5M3 scale) might result in moderate gains compared to FP4, as a consequence to the uniformly-spaced quantization levels provided by the integer representation (in contrast with floating point). However, we have not identified this point of failure in any of the models we investigated, and microscaling FP4 has consistently outperformed INT4 in our studies.
>
> In our experiments throughout the paper, we quantized both weights and activations with microscaling FP4, and applied either the per-tensor scale correction (UE4M3-S) or our novel UE5M3 format _to both equally_. Our discussion ended up focusing primarily on weights, as we identified these tensors to be the main driver for perplexity degradation associated with the narrow-distribution effect, across the majority of models we tested. As a verification, we provide the following ablation study reporting the change in perplexity (the more negative the better) with respect to using FP8 UE4M3 scales, when applying either per-tensor scaling ("per-T") or UE5M3, _separately to weights or activations_. The impact of scaling activations or extending their scale range with UE5M3 in this model is minimal, the perplexity gains are derived almost exclusively from addressing the weights.
>
>
>     | Model           | Block Size | UE4M3 + | UE4M3 + | UE4M3 +   | UE5M3  | UE5M3  | UE5M3  |
>     |                 |            | per-T W | per-T A | per-T W&A | W only | A only | W&A    |
>     | ----------------+------------+---------+---------+-----------+--------+--------+--------|
>     |  llama-3.1-8b   |       8    | -0.431  | -0.012  | -0.470    | -0.425 | -0.013 | -0.445 |
>     |                 |      16    | -0.237  |  0.006  | -0.244    | -0.224 | -0.004 | -0.239 |
>     |                 |      32    | -0.155  | -0.003  | -0.154    | -0.142 | -0.016 | -0.130 |
>     |                 |      64    | -0.081  |  0.014  | -0.075    | -0.072 | -0.001 | -0.080 |
>     |                 |     128    | -0.041  |  0.006  | -0.034    | -0.008 |  0.011 | -0.007 |
>     |                 |     256    | -0.045  |  0.005  | -0.039    |  0.006 | -0.005 | -0.007 |

---

> > ### Author Response · Authors · 2025-11-20
> > **Answer to reviewer bRVj (2)**
> >
> > **2. Demonstrate minimal hardware cost for UE5M3**
> >
> > We thank the reviewer for raising this important question, which gives us the opportunity to expand on the hardware aspects of our exploration. Albeit details of the full hardware implementation are beyond the scope of this paper, we are glad to share the following estimates that allow us to emphasize that UE5M3 is a realistic hardware solution. We have added these details in Appendix K.
> >
> > We synthesized a systolic array processing engine (PE) with a microarchitecture similar to that described in [1]. The engine has eight Single Instruction Multiple Data (SIMD) lanes, and each lane contains multiple multiply-and-accumulate (MAC) engines, each performing MAC operations on multiple weights and input terms, corresponding to different precisions. Each SIMD lane supports BF16, FP8 (both E4M3 and E5M2), INT8, and microscaling FP4. Two versions of microscaling FP4 were synthesized: one with an E4M3 scale and the other with an E5M3 scale.
> >
> > Both E4M3 and E5M3 incur the same multiplier cost for processing the sum of FP4 product terms and the product of the scale mantissas. E5M3 requires a 5-bit adder to compute the product scale exponent, compared to a 4-bit adder for E4M3. The resulting product exponent is further subtracted from the 8-bit exponent of the inter-PE partial sum; therefore, the width of the subsequent adders/datapath remains unchanged.
> >
> > Logic synthesis was performed using a 4 nm process node in a production-grade EDA flow. The area for the E5M3 scale is 0.5 % larger than that for the E4M3 scale, which is negligible and does not affect the bounding box area for place-and-route or subsequent SoC floorplanning. Critical path timing increases by 4 picoseconds, which is negligible for setting the SoC frequency. The intuition behind this small area/timing impact is that the effect of the wider adder is diluted by the arithmetic pipelines for other precisions, as well as non-arithmetic logic such as operand staging and the local register file for operand reuse.
> >
> > [1] Agrawal, A. et al. "9.1 A 7nm 4-core AI chip with 25.6 TFLOPS hybrid FP8 training, 102.4 TOPS INT4 inference and workload-aware throttling", 2021 IEEE International Solid-State Circuits Conference (ISSCC), vol. 64, 144-146, 2021

---

> > > ### Author Response · Authors · 2025-11-20
> > > **Answer to reviewer bRVj (3)**
> > >
> > > **3. Impact of adding a mantissa bit (UE4M4)**
> > >
> > > Repurposing the unused scales bit of FP8 UE4M3 into a UE4M4 format is a valid, alternative approach to the one we propose (UE5M3). On top of increasing precision, the additional mantissa bit has also a _moderate_ effect on the dynamic range: the lowest representable scale value drops from $2^{-9}$ for UE4M3 to $2^{-10}$ for UE4M4 (for comparison, UE5M3 minimum representable value is $2^{-17}$). Based on the findings in our paper, such a decrease is expected to lower the quantization error. We tested the behavior of UE4M4 on granite-3.3-8b and llama-3.1-8b, and included the new results in Appendix J, Fig. 17(a,b). The plots show that UE4M4 is indeed beneficial but UE5M3 remains the superior solution, more effective and robust across various block sizes. We also want to stress that the complexity of scale product mantissa computation increases quadratically with the number of mantissa bits. While the area impact may still be small here due to the presence of logic related to other precisions, increasing the logic depth in the mantissa processing has a direct impact on the critical path, which limits the maximum frequency. In short, for a fixed number of bits per scale, increasing the mantissa width is costlier than increasing the exponent width. This observation also favors UE5M3 as the most hardware-friendly option. We modified the text in Section 5.2 to include a reference to this format and added these new results in Appendix J.
> > >
> > > **4. Motivation for a hardware change in favor of FP8 UE5M3**
> > >
> > > We agree with the reviewer that in several instances FP8 UE4M3 scales with per-tensor scaling achieves comparable accuracy to our proposed UE5M3 format, suggesting that with either strategy the impact of narrow distributions has been effectively (and equivalently) mitigated. However, the motivation behind the introduction of UE5M3 is two-fold. First, robustness: the presence of a single large outlier can hinder the effectiveness of the per-tensor quantization step that is required by UE4M3 to scale up small weights. This is a potential explanation of the gap in perplexity/accuracy in favor of UE5M3, in models where such gap is observed, as in granite-3.3-8b. Second, UE5M3 obviates the need of per-tensor quantization operations. This property of UE5M3 either eliminates the online overheads associated with the dynamic computation of per-tensor scaling factors, or eliminates the need for a data-dependent calibration step leveraged by UE4M3 to estimate such per-tensor scales and apply them statically. We address these motivations in Section 5.1.

---

### Official Review · Reviewer_fiMv · 2025-11-02

**Soundness:** 4
**Presentation:** 3
**Contribution:** 4
**Rating:** 8
**Confidence:** 3

**Summary:**

This paper investigates the unexpected behavior of microscaling FP4 quantization when FP8 scale quantization is used. The authors find that reducing block size does not always reduce error and in fact can worsen accuracy for narrow weight distributions. They develop a theoretical framework that decouples sources of quantization error and show excellent agreement between theory and empirical results across multiple models. Finally, they propose an FP8 UE5M3 scale format that mitigates the anomaly without requiring additional hardware cost, and demonstrate improved model accuracy compared to UE4M3 or per tensor scaling.

**Strengths:**

I appreciate the solid empirical observation and thorough investigation of a subtle but important anomaly in microscaling quantization.

The paper formulates a clear theoretical framework that generalizes the understanding of error behavior and matches experiments well.

The analysis in figures such as Fig 2b and Fig 3c is especially compelling as it isolates the dependence on distribution width and scale quantization.

The proposed UE5M3 solution is simple, hardware friendly, and demonstrates practical effectiveness.

**Weaknesses:**

The anomaly is a surprising phenomenon for readers and it may help to offer a concise intuitive explanation earlier in the introduction, rather than waiting until later sections, so that readers understand the high level mechanism before diving into the detailed framework. For example, a short statement that quantization of scales interacts with narrow distributions and reduces effective representable range could improve clarity.

It would also be valuable to expand the discussion to other scale precisions. The paper focuses on FP8 scales versus FP16 and the new UE5M3 format. Discussion on whether the same anomaly is expected for future lower precision formats such as FP4 scales or mixed mantissa exponent configurations would help generalize the insight.

**Questions:**

Why does FP16 scaling not suffer from this anomaly if the cause is related to the deviation of the maximum weight in a block and scale resolution?

Could you briefly comment on expected behavior for future lower precision scale formats such as FP4 or hybrid exponent mantissa configurations. For example, if microscaling continues to push toward fewer bits for scale, should we expect similar inversion behaviors and would your theoretical framework still apply.

---

> ### Author Response · Authors · 2025-11-20
> **Answer to reviewer fiMv (1)**
>
> We are pleased that the reviewer appreciated our paper and offered thoughtful comments to improve it further. Below, we address the main weaknesses and concerns that were raised.
>
> **1. Add an earlier explanation of the anomaly**
>
> We thank the reviewer for this suggestion. We have altered the Introduction to provide an earlier clarification of the origin of the anomaly:
> "A theoretical framework that decouples sources of quantization errors and explains their interaction with LLM tensor statistics, allowing us to pinpoint the origin of the anomalous behavior to the quantization of the microscaling scales which hinders the representation of low magnitude blocks."
>
> **2. Why do FP16 scales not suffer from the anomaly?**
>
> Throughout our paper, we used _BF16_ scales (see Fig. 1a and 3c) as the counterpart to the FP8 E4M3 scales that are used in conventional microscaling formats. BF16, being a 1-8-7 format, provides wide dynamic range and high precision representation of the scaling factors. Hence, perplexity inversion does not materialize. We selected BF16 because this is the standard format for pre- and post-conversion for microscaling formats (see for example Microsoft implementation of MX at [1]).
>
> Although we haven't tested IEEE FP16 (1-5-10) scales, it is reasonable to expect this format would not show perplexity inversion either. This is in virtue of being a much more precise version of our proposed UE5M3, with a minimum representable value as low as $2^{-24}$.
>
> We stress however that 16-bit formats for scaling factors have a significant impact on hardware complexity, which motivated the industry to adopt 8-bit formats for microscaling quantization.
>
> [1] https://github.com/microsoft/microxcaling?tab=readme-ov-file\#spec-configuration

---

> > ### Author Response · Authors · 2025-11-20
> > **Answer to reviewer fiMv (2)**
> >
> > **3. Expand discussion to other scale precisions**
> >
> > We agree with the reviewer that generalizing the investigation to other quantization formats would strengthen the paper further.
> >
> > First, we'd like to mention that we have expanded Appendix G, supplementing the earlier analysis based on theoretical predictions and synthetic data of _microscaling INT4 with FP8 UE4M3 scales_, with measurements of perplexity across various LLM and hybrid SSM. To collect these data, we used the same methodology we employed across the paper for FP4 quantization, and compared results for UE4M3, UE4M3-S, and UE5M3 scales.
> >
> > Importantly, perplexity inversion is still observed across these models, albeit in a less pronounced manner. Both UE4M3-S and UE5M3 are effective in mitigating the inversion and improving performance, with UE5M3 achieving comparable or better perplexity than UE4M3-S.
> >
> > These results are consistent with the analysis of ideal normal distributions (synthetic data) as well as our theoretical framework (Fig. 13 in Appendix G): both show that narrow element distributions quantized as INT4 do experience a qualitatively similar effect as FP4, with smaller block size producing larger error than larger block size in narrow distributions. For block size 16 vs. 8, for example, the crossover does occur at a lower standard deviation ($\approx 1.5 \cdot 10^{-2}$) than for FP4 ($\approx 2 \cdot 10^{-2}$), suggesting that considering all tensors within a model, and their spread of standard deviations, we should observe a less pronounced effect, possibly arising at lower block sizes than FP4, at which the increase in error is larger. This is indeed what Fig. 13 and the newly added Fig. 14 show.
> >
> > In the context of more aggressive formats for _scales_ quantization, we have now measured the impact of microscaling FP4 with two custom FP6 formats and added these results in Appendix H. We stress that no standard format has been recommended for FP6 _scales_ at this time, and while OCP [1] prescribes FP6 E3M2 and FP6 E2M3 for _elements_ quantization in MXFP6 formats, based on the findings in our paper we believe supporting the widest dynamic range in scales is of utmost importance. Hence, we tested FP6 UE5M1 and FP6 UE4M2 scales (assuming repurposing of the unused sign bit), in combination with FP4 E2M1 elements, to quantize weights and activations of llama-3.1-8b (BF16 baseline perplexity = 6.242). Consistently with UE4M3-S nomenclature, UE5M1-S and UE4M2-S represent the use of per-tensor scale on both weights and activations.
> >
> >     | block size |  UE5M1  | UE5M1-S |   UE4M2   | UE4M2-S |
> >     |------------+---------+---------+-----------+---------|
> >     |      2     |  7.217  |  7.240  | 46795.660 |  6.598  |
> >     |      4     |  7.265  |  7.280  |   343.621 |  6.726  |
> >     |      8     |  7.316  |  7.346  |    19.641 |  6.872  |
> >     |     16     |  7.414  |  7.424  |    11.398 |  7.038  |
> >     |     32     |  7.519  |  7.547  |     9.348 |  7.182  |
> >     |     64     |  7.715  |  7.696  |     9.607 |  7.385  |
> >     |    128     |  7.955  |  7.952  |    10.955 |  7.660  |
> >     |    256     |  8.328  |  8.307  |     9.017 |  8.019  |
> >
> > Results are entirely consistent with our previous conclusions related to FP8 scales: the wider dynamic range provided by UE5M1 achieves limited degradation in absence of a per-tensor scale. In fact, applying such global scale has negligible effect on perplexity when using this format. However, compared to UE5M3, this format suffers from the loss of 2 bits of precision. On the other hand, UE4M2 performs very poorly without per-tensor scale and we once again we observe a _perplexity inversion_ behavior. This can be effectively mitigated by means of a per-tensor scale (UE4M2-S).
> >
> > With respect to the validation of our theoretical framework, we modeled these two microscaling formats based on FP4 E2M1 elements and FP6 scales, either UE5M1 and UE4M2. MSE estimates are reported in Fig. 15(a,b) (Appendix H) across different block sizes (bs). In the case of FP6 UE5M1 scales, no crossover is observed between curves of different block size, consistently with absence of perplexity inversion reported above. By contrast, MSE estimates for FP6 UE4M2 show similar features as FP8 E4M3 (as in Fig. 3c and Fig. 11), but accentuated by the moderate decrease in dynamic range and lower precision brought about by the smaller mantissa. Interestingly, for FP6 UE4M2, crossover occurs at larger $\sigma$ compared to FP8 E4M3 (for example, at $\sigma \approx 3.8 \cdot 10^{-2}$ going from block size 16 to 8, compared to $\sigma \approx 2 \cdot 10^{-2}$ we reported in Sec.~3.2), which suggests even wider distributions, and hence more models, would experience larger error at smaller block sizes. Hence, mitigation becomes even more important as more aggressive quantization formats are deployed.
> >
> > [1] Rouhani et al., "OCP microscaling formats (mx) specification version 1.0" Technical report, Open Compute Project OCP, 2023

---

### Official Review · Reviewer_qJkV · 2025-11-02

**Soundness:** 3
**Presentation:** 3
**Contribution:** 3
**Rating:** 6
**Confidence:** 4

**Summary:**

The paper investigates an unexpected limitation of microscaling quantization—a fine-grained, per-block quantization method increasingly used for efficient training and inference of large language models (LLMs). While smaller block sizes are generally assumed to improve quantization accuracy, the authors discover a quantization anomaly whereby further reducing block size below a threshold increases model perplexity, a phenomenon they term perplexity inversion. They diagnose this anomaly through extensive experimentation across various LLMs and develop a robust theoretical framework that decomposes the Mean Squared Error (MSE) into three distinct contributions, revealing that the quantization of scaling factors and the effect of zero-rounding are the primary drivers of the inversion, especially for narrow tensor distributions. To address this, the paper proposes FP8 unsigned E5M3 scales, demonstrating that this hardware-friendly solution effectively mitigates perplexity inversion by offering an increased dynamic range without the need for global scaling operations.

**Strengths:**

1. The paper makes a highly original and significant contribution by identifying "perplexity inversion," a counter-intuitive phenomenon where smaller block sizes unexpectedly increase quantization error in microscaling for LLMs. This challenges common assumptions and reveals a critical pitfall for future low-bit quantization efforts.
2. The FP8 UE5M3 solution stands out as a key strength due to its practical and well-reasoned approach to mitigating the identified perplexity inversion. By repurposing an unused bit to extend the exponent range, UE5M3 significantly increases dynamic range, enabling better representation of small-magnitude elements crucial for narrow tensor distributions. This design is not only hardware-friendly, requiring minimal modifications to existing infrastructure, but also achieves comparable or superior performance to more complex per-tensor scaling methods, effectively simplifying the quantization pipeline while preserving or enhancing model accuracy. Its foundation in the paper's theoretical insights ensures it is a targeted and robust solution to the core problem.

**Weaknesses:**

1. The paper effectively shows perplexity inversion with FP4 elements and FP8 UE4M3 scales. Do the authors observe similar inversion with other low-bit formats (e.g., INT4, INT8, other FP formats) and quantized scales? Clarifying if this mechanism is universally applicable or specific to the studied configuration would define the discovery's scope.
2. The paper emphasizes the hardware-friendly nature of UE5M3, particularly for inference. However, the practical implications of integrating UE5M3 during the training phase, especially for quantization-aware training (QAT), are not fully elaborated. The discussion mainly focuses on FP8 UE4M3 for existing hardware. Clarifying how the extended exponent range of UE5M3 impacts gradient calculations, potential numerical stability issues during training, or if it primarily targets post-training quantization (PTQ) scenarios would be beneficial.
3. While the proposed FP8-UE5M3 format effectively extends the dynamic range of fixed scales, the paper lacks a comparison with adaptive scaling methods such as VS-Quant (Per-vector Scaled Quantization for Accurate Low-Precision Neural Network Inference) and GWQ (Gradient-Aware Weight Quantization for Large Language Models). These approaches also refine scale granularity through per-vector or group-wise learnable scaling. Without an empirical or qualitative comparison, it remains unclear whether UE5M3 offers distinct advantages over these adaptive strategies or if their benefits overlap.

**Questions:**

1. The paper effectively demonstrates perplexity inversion using FP4 elements and FP8 UE4M3 scales. It would be valuable to understand if this phenomenon is specific to this combination or broadly applicable. Have the authors observed similar inversion when using other common low-bit quantization formats (e.g., INT4, INT8, or different FP variants) with their corresponding scales also quantized? Clarifying whether the identified mechanism (interplay of narrow distributions and limited scale dynamic range) is a universal challenge or uniquely pronounced with the studied configuration would better define the scope and novelty of this important discovery.
2. The benefits of UE5M3 are predominantly highlighted for inference. However, its role in the training phase, particularly within a Quantization-Aware Training (QAT) framework, needs further clarification. Could the authors elaborate on how the extended exponent range of UE5M3 impacts gradient calculations, numerical stability, or convergence during QAT compared to UE4M3? Understanding whether UE5M3 is primarily geared towards Post-Training Quantization (PTQ) or if it seamlessly integrates with QAT, potentially requiring specific modifications or offering distinct advantages, would provide a more complete view of its practical utility.
3. While the UE5M3 format improves scale representation through a fixed hardware-defined design, it would be valuable to compare it with adaptive scaling methods such as VS-Quant (Dai et al., 2021) and GWQ (Yang et al., 2024), which optimize scales per vector or group. These methods share the goal of mitigating quantization errors via finer-grained scaling. An empirical or qualitative comparison discussing complexity, overhead, and accuracy trade-offs would help clarify UE5M3’s unique advantages and positioning within the current quantization landscape.

---

> ### Author Response · Authors · 2025-11-20
> **Answer to reviewer qJkV (1)**
>
> We thank the reviewer for the careful evaluation of our paper and the constructive comments offered in their review. Below, we address the main identified weakness and concerns.
>
> **1. Does inversion appear in other microscaling formats?**
>
> Yes, we do observe inversion with other microscaling formats using quantized scales. We have now added Fig. 14 in Appendix G to supplement the theoretical analysis with measurements of perplexity for an INT4 element-quantization format using FP8 scales. To collect these data, we used the same methodology we employed throughout the paper for FP4 quantization, and compared results for UE4M3, UE4M3-S, and UE5M3 scales.
>
> Importantly, perplexity inversion is still observed across these models, albeit in a less pronounced manner. Both UE4M3-S and UE5M3 are effective in mitigating the inversion and improving performance, with UE5M3 achieving comparable or better perplexity than UE4M3-S.
>
> These results are consistent with the analysis of ideal normal distributions (synthetic data) as well as our theoretical framework (Fig. 13 in Appendix G): both show that narrow element distributions quantized as INT4 do experience a qualitatively similar effect as FP4, with smaller block size producing larger error than larger block size in narrow distributions. The crossover does occur at a lower standard deviation ($\approx 1.5 \cdot 10^{-2}$) than for FP4 ($\approx 2 \cdot 10^{-2}$), suggesting that considering all tensors within a model, and their spread of standard deviations, we should observe a less pronounced effect, possibly arising at lower block sizes than FP4, at which the increase in error is larger. This is indeed what Fig. 13 and the newly added Fig. 14 show.
>
> In the context of alternative microscaling formats with more aggressive _scales_ quantization, we have now measured the impact of microscaling FP4 with two custom FP6 formats and added these results in Appendix H. We stress that no standard format has been recommended for FP6 _scales_ at this time, and while OCP [1] prescribes FP6 E3M2 and FP6 E2M3 for _elements_ quantization in MXFP6 formats, based on the findings in our paper we believe supporting the widest dynamic range in scales is of utmost importance. Hence, we tested FP6 UE5M1 and FP6 UE4M2 scales (assuming repurposing of the unused sign bit), in combination with FP4 E2M1 elements, to quantize weights and activations of llama-3.1-8b (BF16 baseline perplexity = 6.242). Consistently with UE4M3-S nomenclature, UE5M1-S and UE4M2-S represent the use of per-tensor scale on both weights and activations.
>
>     | block size |  UE5M1  | UE5M1-S |   UE4M2   | UE4M2-S |
>     |------------+---------+---------+-----------+---------|
>     |      2     |  7.217  |  7.240  | 46795.660 |  6.598  |
>     |      4     |  7.265  |  7.280  |   343.621 |  6.726  |
>     |      8     |  7.316  |  7.346  |    19.641 |  6.872  |
>     |     16     |  7.414  |  7.424  |    11.398 |  7.038  |
>     |     32     |  7.519  |  7.547  |     9.348 |  7.182  |
>     |     64     |  7.715  |  7.696  |     9.607 |  7.385  |
>     |    128     |  7.955  |  7.952  |    10.955 |  7.660  |
>     |    256     |  8.328  |  8.307  |     9.017 |  8.019  |
>
> Results are entirely consistent with our previous conclusions related to FP8 scales: the wider dynamic range provided by UE5M1 achieves limited degradation in absence of a per-tensor scale. In fact, applying such global scale has negligible effect on perplexity when using this format. However, compared to UE5M3, this format suffers from the loss of 2 bits of precision. On the other hand, UE4M2 performs very poorly without per-tensor scale and we once again we observe a _perplexity inversion_ behavior. This can be effectively mitigated by means of a per-tensor scale (UE4M2-S)
>
> Applying our theoretical framework to this new scenario (see Fig. 15(a,b) in Appendix H) provides predictions in agreement with the observed behavior of these two custom microscaling FP4 formats with FP6 scales.

---

> > ### Author Response · Authors · 2025-11-20
> > **Answer to reviewer qJkV (2)**
> >
> > **2. Implication for training**
> >
> > We thank the reviewer for raising the question of how UE5M3 interacts with training and quantization-aware training (QAT). While our experiments focus on inference, the underlying mechanism we identify, the increased error at small block sizes caused by scale quantization, occurs at the tensor level, not only at the model level. Because the anomaly originates from the interaction between narrow weight and activation distributions with the limited dynamic range of UE4M3 scales, the same effect necessarily manifests during training whenever parameters or activations have small magnitude and naive UE4M3 scales are used.
> >
> > In this regard, UE5M3 is not tied to post-training quantization (PTQ). Its extended exponent range directly improves the fidelity of block-wise scales during training, in both forward and backward passes, by reducing the probability of scale underflow and eliminating the “all-zero block” failure mode, and associated quantization errors. This reduces gradient corruption due to inaccurate rescaling and thus improves numerical stability. UE4M3-S (with per-tensor scaling) achieves a similar effect, though with additional overhead and sensitivity to outliers, as outlined in Section~5.1. On the other hand, UE5M3 provides a hardware-friendly alternative that mitigates the same source of error without global scaling operations.
> >
> > Regarding gradient calculations: because scale quantization errors propagate multiplicatively into both weight and activation gradients, increasing the representable range of the scale directly reduces the variance of gradient noise introduced by quantization. While our study does not yet evaluate full QAT pipelines, the mechanism analyzed in Sections 3 and 4 applies symmetrically: UE5M3 mitigates the principal source of tensor-level error during training just as it does at inference.
> >
> > A full empirical investigation of training-time behavior is a promising direction for future work, which we are actively exploring.
> >
> > **3. Comparison to adaptive strategies**
> >
> > We appreciate the suggestion to position our proposed UE5M3 format relative to adaptive scaling methods. These approaches do share the broad goal of mitigating quantization error through improved scale selection, but they differ fundamentally in their implementation, as well as associated computational and hardware costs.
> >
> > Our work focuses specifically on microscaling formats, where quantization granularity is already at very small block sizes (e.g., 8 to 32), and where, as demonstrated in the paper, the dominant source of error is the limited dynamic range of the scale format. In this regime, the anomaly we identify is not due to suboptimal scale placement, but to the inability of UE4M3 to represent sufficiently small scales. UE5M3 directly resolves this failure mode by extending the exponent range without increasing mantissa cost, nor requiring additional per-block computations, nor calibration steps.
> >
> > In contrast, adaptive strategies such as those mentioned introduce per-vector or per-group scale computation, involving Hessian- or gradient-driven optimization or additional calibration steps. In principle, such strategies may help determining a more appropriate quantization range, _when coupled with a process that pushes tensor distributions into a more quantizable range_. Simply altering the quantization boundaries (clips) cannot eliminate the quantization error associated with small-magnitude blocks, which already utilize, when using absmax, the smallest representable scaling factor permitted by a given data format. We note, however, that such retraining methods require additional compute budget, necessitate the use of calibration datasets and, to the best of our knowledge, they have not yet been studied thus far in the context of microscaling quantization. We view learned clip optimization for microscaling formats as an interesting avenue of investigation, one that however may not necessarily bypass the representation limit of UE4M3. UE5M3 resolves this limit at the hardware level with minimal additional complexity, allowing microscaling to function robustly even for very narrow weight and activation distributions.

---

### Author Response · Authors · 2025-11-20
**revised manuscript**

We sincerely thank all reviewers for their attentive evaluation of our paper and for providing valuable feedback. We have uploaded a revised version of the manuscript, with changes to the text highlighted in blue. The revision also includes major changes to the appendices. We remain available to answer additional questions and provide further clarifications.

---

### Comment · Area_Chair_2XBG · 2025-11-20
**Action Needed: Review Rebuttal and Update Evaluation**

Dear Reviewers,

Thank you, as always, for your valuable contributions and efforts. The authors have now submitted their rebuttal. Please take a moment to review it and provide any necessary follow-up actions, such as additional questions, clarification requests, or updates to your review.

Since the initial ratings ranged from 2 to 8, I kindly ask you to pay close attention to the perspectives of the other reviewers when preparing your final response.

Thank you again for your support.

---

### Author Response · Authors · 2025-12-03
**Overview**

We thank again the reviewers for their thoughtful feedback. Since recent circumstances prevent them from further engaging or updating their scores, we would like to highlight several points that strongly support acceptance.

1. **Novel and impactful contribution**: reviewers consistently praised the originality and significant of our findings in understanding microscaling quantization for LLM (qJkV: "highly original and significant contribution", UtsX: "perplexity inversion phenomenon is novel and well-motivated"). The discovery of perplexity inversion is repeatedly emphasized as an important, counter-intuitive result that challenges long-held assumptions about fine-grained quantization.

2. **Strong and general theoretical framework**: multiple reviewers commended the rigor, clarity, and empirical accuracy of our theoretical analysis (UtsX: "The theoretical modeling is rigorous and matches experimental data convincingly", bRVj: "mathematical framework that perfectly explains the why behind the anomaly", fiMv: "clear theoretical framework that generalizes the understanding of error behavior and matches experiments well"). They noted that our model cleanly decomposes the sources of quantization error, explains the anomaly precisely, and aligns closely with extensive experimental validation.

3. **Effective and hardware-friendly solution**: the proposed FP8 UE5M3 scale format was widely recognized as a simple, practical, and impactful fix (UtsX: "The proposed UE5M3 solution is simple, practical, and hardware-friendly", fiMv: "The proposed UE5M3 solution is simple, hardware friendly, and demonstrates practical effectiveness", qJkV: "The FP8 UE5M3 solution stands out as a key strength due to its practical and well-reasoned approach").

4. **Clear presentation and comprehensive experiments**: reviewers also remarked that the paper is well written, clearly presented, and supported by robust experiments, insightful figures, and thorough investigation across models and settings.

5. **Key additional improvements** made in response to reviewers' questions:
	- we expanded our analysis beyond microscaling FP4 and validated our theoretical framework for INT4 elements and FP6 scales, supporting its generality for future, more aggressive quantization schemes
	- we demonstrated that perplexity inversion persists in larger LLMs, up to 70B parameters
	- we provided quantitative hardware feasibility estimates for FP8 UE5M3, including area, timing, and details on a real-world on-chip implementation

6. More specifically, in **response to the concerns of reviewer bRVj**, who gave our paper a low initial score, we addressed the following weaknesses (W) and questions (Q):
	- W1/Q1: we showed that asymmetric activation distributions would likely benefit from FP8 UE5M3 scales and demonstrated that in the models we tested weights were the primary driver of perplexity inversion
	- W2: we provided quantitative area and timings estimates of hardware feasibility of UE5M3 (added in Appendix K)
	- Q2: we tested UE4M4 format (added in Appendix J) but UE5M3 remains the more robust and hardware-friendly option
	- Q3: we clarified the motivations in support of FP8 UE5M3 scales introduction

Overall, we appreciate that the reviewers viewed the work as delivering an original empirical discovery, a rigorous theoretical explanation, and a practical, hardware-efficient solution. Finally, we thank the Area Chair for their time and effort in reviewing this submission.

---

### Meta-Review · Area_Chair_qzG9 · 2025-12-17

**Summary:**

This submission presents an original and impactful study on microscaling quantization for LLMs, centered on the discovery of the “perplexity inversion” phenomenon. Reviewers consistently praised the novelty and significance of this empirical finding, noting that it challenges prevailing assumptions about the benefits of ever finer-grained quantization. The paper couples this discovery with a clear, rigorous theoretical framework that decomposes quantization error sources and matches empirical behavior across models and settings. Multiple reviews highlighted the strength of the modeling and its close alignment with experimental results.

The proposed FP8 UE5M3 scaling format is a practical, hardware-friendly solution. The authors provide quantitative estimates of area and timing, as well as details for an on-chip implementation, which strengthens the case for feasibility. The paper is clearly written and supported by comprehensive experiments, insightful figures, and careful analysis.

The authors further improved the work during rebuttal: they validated the theory beyond microscaling FP4 by testing INT4 elements and FP6 scales, and showed that perplexity inversion persists in larger LLMs up to 70B parameters. They also addressed specific concerns from a critical review (bRVj): (1) they demonstrated that weights are the primary driver of perplexity inversion and argued that asymmetric activations would likely benefit from FP8 UE5M3; (2) they provided quantitative hardware feasibility estimates (Appendix K); (3) they evaluated UE4M4 (Appendix J) but concluded UE5M3 is more robust and hardware-friendly; and (4) they clarified the motivation for introducing FP8 UE5M3 scales.

Overall, the paper delivers a coherent package: an original empirical discovery, a convincing theoretical explanation, and a practical mitigation strategy. Minor suggestions include broadening tasks and datasets, releasing code and calibration pipelines to facilitate reproducibility, and expanding real-world deployment evidence (e.g., energy/latency trade-offs under different block sizes and scaling granularities). These do not detract from the core contribution.

Recommendation: Accept

**Reviewer Concerns:**

1. Reviewer qJkV (mostly solved)
2. Reviewer fiMv (mostly solved)
3. Reviewer bRVj (partly solved)
4. Reviewer UtsX (mostly solved)

**Reviewer Scores:**

1. Reviewer qJkV would keep score 6
2. Reviewer fiMv would keep score 8
3. Reviewer bRVj would increase the score to 4 or 6
4. Reviewer UtsX (mostly solved) would keep the score 6.

---

### Decision · Program_Chairs · 2026-01-26

Accept (Poster)